


# Carbon stocks and fluxes in the high latitudes: Using site-level data to evaluate Earth system models

Sarah Chadburn[1,2], Gerhard Krinner[3], Philipp Porada[4,5], Annett Bartsch[6,7], Christian Beer[4,5], Luca Belelli Marchesini[8,9], Julia Boike[10], Bo Elberling[11], Thomas Friborg[11], Gustaf Hugelius[12], Margareta Johansson[13], Peter Kuhry[12], Lars Kutzbach[14], Moritz Langer[10], Magnus Lund[15], Frans-Jan Parmentier[16], Shushi Peng[3,17], Ko Van Huissteden[9], Tao Wang[18], Sebastian Westermann[19], Dan Zhu[20], and Eleanor Burke[21]

[1]University of Leeds, School of Earth and Environment, Leeds LS2 9JT, U.K.
[2]University of Exeter, College of Engineering, Mathematics and Physical sciences, Exeter EX4 4QF, U.K.
[3]Laboratoire de Glaciologie et Géophysique de l'Environnement (LGGE), 38041 Grenoble, France
[4]Department of Environmental Science and Analytical Chemistry, Stockholm University, 10691 Stockholm, Sweden
[5]Bolin Centre for Climate Research, Stockholm University
[6]Vienna University of Technology, Vienna, Austria
[7]Austrian Polar Research Institute, Vienna, Austria
[8]School of Natural Sciences, Far Eastern Federal University, Vladivostok (Russia)
[9]Department of Earth Sciences, Vrije Universiteit (VU) Amsterdam, The Netherlands
[10]Alfred Wegener Institute, Helmholtz Center for Polar and Marine Research (AWI) 14473 Potsdam, Germany
[11]Center for Permafrost (CENPERM), Department of Geosciences and Natural Resource Management, University of Copenhagen, Denmark
[12]Department of Physical Geography, Stockholm University, 10691 Stockholm, Sweden
[13]Dept. of Physical Geography and Ecosystem, Lund University, Sölvegatan 12, 223 62 Lund, Sweden
[14]Institute of Soil Science, Center for Earth System Research and Sustainability, Universität Hamburg, Hamburg, Germany
[15]Department of Bioscience, Arctic Research Center, Aarhus University, Frederiksborgvej 399, DK-4000 Roskilde, Denmark
[16]Department of Arctic and Marine Biology, UiT - The Arctic University of Norway, Tromsø, Norway
[17]Sino-French Institute for Earth System Science, College of Urban and Environmental Sciences, Peking University, Beijing 100871, China
[18]Key Laboratory of Alpine Ecology and Biodiversity, Institute of Tibetan Plateau Research and Center for Excellence in Tibetan Plateau Earth Sciences, Chinese Academy of Sciences, Beijing 100085, China
[19]Univesity of Oslo, Department of Geosciences, P.O. Box 1047 Blindern, NO-0316 Oslo, Norway
[20]Laboratoire des Sciences du Climat et de l'Environnement, LSCE CEA CNRS UVSQ, Gif Sur Yvette, France
[21]Met Office Hadley Centre, Fitzroy Road, Exeter EX1 3PB, U.K.

*Correspondence to:* Sarah Chadburn (s.e.chadburn@exeter.ac.uk)



**Abstract.**

It is important that climate models can accurately simulate the terrestrial carbon cycle in the Arctic, due to the large and potentially labile carbon stocks found in permafrost-affected environments, which can lead to a positive climate feedback, along with the possibility of future carbon sinks from

northward expansion of vegetation under climate warming. Here we evaluate the simulation of tundra carbon stocks and fluxes in three land surface schemes that each form part of major Earth System Models (JSBACH, Germany; JULES, UK and ORCHIDEE, France). We use a site-level approach where comprehensive, high-frequency datasets allow us to disentangle the importance of different processes. The models have improved physical permafrost processes and there is a reasonable corre-

spondence between the simulated and measured physical variables, including soil temperature, soil moisture and snow.

We show that if the models simulate the correct leaf area index (LAI), the standard C3 photosynthesis schemes produce the correct order of magnitude of carbon fluxes. Therefore, simulating the correct LAI is one of the first priorities. LAI depends quite strongly on climatic variables alone, as

we see by the fact that the dynamic vegetation model can simulate most of the differences in LAI between sites, based almost entirely on climate inputs. However, we also identify an influence from nutrient limitation as the LAI becomes too large at some of the more nutrient-limited sites. We conclude that including moss as well as vascular plants is of primary importance to the carbon budget, as moss contributes a large fraction to the seasonal $CO_2$ flux in nutrient-limited conditions. Moss

photosynthetic activity can be strongly influenced by the moisture content of moss, and the carbon uptake can be significantly different from vascular plants with similar LAI.

The soil carbon stocks depend strongly on the rate of input of carbon from the vegetation to the soil, and our analysis suggests that an improved simulation of photosynthesis would also lead to an improved simulation of soil carbon stocks. However, the stocks are also influenced by soil

carbon burial (e.g. through cryoturbation) and the rate of heterotrophic respiration, which depends on the soil physical state. More detailed below-ground measurements are needed to fully evaluate soil biological and physical processes. Furthermore, even if these processes are well modelled, the soil carbon profiles cannot resemble peat layers as peat accumulation processes are not represented in the models.

Thus we identify three priority areas for model development: 1. Dynamic vegetation including a. climate and b. nutrient limitation effects. 2. Adding moss as a plant functional type. 3. Improved vertical profile of soil carbon including peat processes.

## 1  Introduction

Land areas in northern high latitudes may represent a net source or a net sink of carbon to the

atmosphere in the future, and there is not yet a consensus as to which of the two is more likely, e.g.



(Cahoon et al., 2012; Hayes et al., 2011). This is not because it is likely to be small: on a pan-Arctic scale we could see anything between a net emission of over 100GtC or a net sink of up to 60GtC by the end of this century (Schuur et al., 2015; Qian et al., 2010). To put this into context, the remaining emissions budget in order to stabilise climate warming below 2°C above pre-industrial levels is less

than 250GtC from 2017 (Peters et al., 2015), so it is very important to reduce uncertainty in the northern high latitude carbon cycle. The uncertainty comes largely from the representation of these processes in Earth System Models (ESM's), which are our main tool for future climate projections.

The potential for large carbon emissions comes from the large quantities of old carbon that are frozen into permafrost, protected from decomposition under the current cold climate. Around 800Gt

of carbon is stored in permanently frozen soils (Hugelius et al., 2014). If the permafrost thaws, this carbon may decompose and be released to the atmosphere (Burke et al., 2012, 2013; Koven et al., 2015; Schneider von Deimling et al., 2012, 2015; MacDougall and Knutti, 2016). On the other hand, the increased vegetation growth that is already taking place in the Arctic under climate warming (Tucker et al., 2001; Tape et al., 2006) could result in a net uptake of carbon from the atmosphere

(Quegan et al., 2011; Qian et al., 2010). It should be noted, however, that in some areas Arctic vegetation growth is not increasing but rather 'browning' (Epstein et al., 2016).

The representations of both permafrost carbon and Arctic vegetation in Earth System Models are not well developed. Some models now include a vertical representation of soil carbon which allows the frozen carbon in permafrost to be included (Koven et al., 2009, 2013; Schaphoff et al., 2013;

Burke et al., 2017), but most do not yet represent important mechanisms of carbon storage and release, such as sedimentation, thermokarst formation, and a proper representation of cryoturbation (Schneider von Deimling et al., 2015; Beer, 2016), although sedimentation is included in Zhu et al. (2016). There is also a growing consensus that the chemical decomposition models used in ESMs are not adequate to represent microbial processes (Wieder et al., 2013; Xenakis and Williams, 2014).

Vegetation models also, for the most part, do not include the appropriate high latitude vegetation types and those models that have dynamic vegetation are lacking in processes that are essential determinants of vegetation dynamics, such as nutrient limitation and interactions with soil (Wieder et al., 2015).

In this paper we assess the ability of the land surface components from three Earth System Models

to represent the observed carbon stocks and fluxes at tundra sites, identifying the processes that have the greatest impact on the uncertainty. These processes are therefore priorities for future model development.

This is a synthesis from the recently concluded EU project PAGE21 (Permafrost in the Arctic and Global Effects in the 21st century), evaluating the models that took part in the project (described

in Section 2, below) at the five PAGE21 primary sites, which are all located in Arctic permafrost regions, specifically Siberia, Sweden, Svalbard and Greenland. After the site-level evaluation of

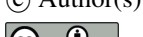



physical processes by Ekici et al. (2015), this evaluation of carbon cycle processes continues site-level model evaluation efforts. The sites are described in detail in Section 3.

## 2   Model descriptions

The three models studied here are JSBACH, JULES and ORCHIDEE. These are all land surface components of major Earth System Models. They can be run in a coupled mode within the ESM, or, as here, they can be run standalone forced by observed meteorology. Each model had some development of high latitude processes during the PAGE21 project, and model developments have also been ongoing since the conclusion of the project in late 2015 (see below).

### 2.1   JSBACH

The Jena Scheme for Biosphere-Atmosphere Coupling in Hamburg (JSBACH 3.0 (Raddatz et al., 2007; Brovkin et al., 2009)) is the land surface component of the Max Planck Institute Earth system model (MPI-ESM). The model simulates water fluxes, heat fluxes, and carbon fluxes from vegetation and soil via one-dimensional vertical fluxes. Photosynthesis in JSBACH is based on the approaches

of Farquhar et al. (1980) and Collatz et al. (1992), as described in Knorr (2000). The carbon cycle is represented by three vegetation pools (active, reserves, wood) and five soil carbon pools which are defined by solubility (Goll et al., 2015). However, the soil carbon model does not have a vertical dimension.

Hydrological fluxes are simulated by a five-layer scheme (Hagemann and Stacke, 2015). The

model is run as a gridded set of points for large scale simulations. Each grid cell is subdivided into tiles which represent different vegetation types and which can vary in fractional cover. During PAGE21, soil freezing, dynamic snow layers and a simple organic layer were added in JSBACH (Ekici et al., 2014). In the version used in this paper, the simple organic layer is switched off and replaced by a moss layer with dynamic soil moisture contents and thermal properties (Porada et al.,

2016), and additional soil layers were added in order to represent a 50 m depth. The moss carbon fluxes (photosynthesis, respiration) are also simulated, as in the model described by Porada et al. (2013). In the version used here, the moss carbon fluxes are not yet fully coupled into the JSBACH carbon cycle, so the moss carbon fluxes are considered separately in the analysis that follows.

### 2.2   JULES

JULES is the land surface component of the new community Earth System model, UKESM (Jones and Sellar, 2015). It can also be run offline forced by observed meteorology, and it can be run at a regional or point scale as well as globally. JULES is described in Best et al. (2011); Clark et al. (2011). It is a community model with many users and many ongoing developments. JULES includes a dynamic vegetation model (TRIFFID), surface energy balance, a dynamic snowpack model (ver-



tical processes only), vertical heat and water fluxes, soil freezing, large scale hydrology, and carbon
fluxes and storage in both vegetation and soil. It also includes specific representations of crops, urban
heat and water dynamics, fire diagnostics and river routing.

During PAGE21 the permafrost physics in JULES was improved (Chadburn et al., 2015a), and a
vertical representation of soil carbon, including cryotubation mixing, was added (Burke et al., 2017).

In this work the vertical soil carbon, organic soil properties, deep soil column (including bedrock)
and high resolution soil are used. We also use the 9 PFT's described in Harper et al. (2016) and the
latest set of PFT parameters from the UKESM project. For more details of soil and vegetation con-
figuration see Simulation Set-up (Section 4.2) and Appendix. The version of JULES used is available
on https://code.metoffice.gov.uk/svn/jules/main/branches/dev/eleanorburke/vn4.3_permafrost.

## 2.3 ORCHIDEE

ORCHIDEE is the land-surface component of the IPSL climate model as well as a standalone land
surface model. ORCHIDEE simulates the principal processes of the biosphere influencing the global
carbon cycle (photosynthesis, autotrophic and heterotrophic respiration of plants and in soils, fire,
etc.) as well as latent, sensible, and kinetic energy exchanges at the land surface (Krinner et al.,

120  2005).

The ORCHIDEE high-latitude version includes vertically resolved soil carbon and cryoturbative
mixing (Koven et al., 2009), a scheme describing soil freezing and its effect on soil thermal and
hydrological dynamics (Gouttevin et al., 2012), and a multi-layer snow scheme with improved rep-
resentation of snow thermal conductivity, as well as snow settling, water percolation and refreezing

(Wang et al., 2013). In its latest version used in this study, the impacts of soil organic matter on soil
thermal and hydraulic properties, including porosity, thermal conductivity, heat capacity and water
holding capacity, are incorporated in the model, generally following Lawrence and Slater (2008).
The observation-based soil organic carbon map from NCSCD (Hugelius et al., 2014) is used in the
thermal and hydrological modules to derive the above mentioned soil properties, after linear interpo-

lation from their original 4-layer (i.e. 0-30, 30-100, 100-200, 200-300 cm) values to fit ORCHIDEE
vertical layers. The latest ORCHIDEE now has the same vertical discretization scheme for the ther-
mal and hydrological modules above 2 m (11 layers), while the thermal module further extends to
38 m (total 32 layers). ORCHIDEE has 13 PFT's, but there is no specific high-latitude PFT in the
version used here, so C3 grasses are prescribed as a fixed land cover (but with dynamic phenology).

## 3  Site descriptions

The sites represent a range of climatological and biogeophysical conditions across the tundra. Abisko
is the warmest site, in the sporadic permafrost zone, followed by Bayelva, which is a high Arctic
maritime site (on Svalbard), and Zackenberg, which is a maritime site in Greenland (colder than



Bayelva). Samoylov and Kytalyk have a continental Siberian climate and the coldest mean annual
temperatures. The soil types, vegetation types and the wetness of the ground all vary between sites.
The landscapes at each site also differ, which can influence the permafrost and carbon dynamics,
for example via wind-blown snow and lateral water fluxes. The following sections provide a short
description of each study area, and the important climatic and permafrost variables are given in Table
1.

## 3.1 Abisko

The Abisko site (68°21' N, 18°49' E, 385m a.s.l) is located about 200 km north of the Arctic Circle
in the Torneträsk catchment, northernmost Sweden. The catchment ranges from 345 m a.s.l. to 1700
m a.s.l. and is centered around Lake Torneträsk. Mean annual air temperature is close to 0°C (-0.6°C
for the period 1913-2006), and warming has resulted in mean annual air temperatures above 0°C for
the last decade (Callaghan et al. (2010); Abisko Station meteorological data; www.polar.se/abisko).
The Abisko area is situated in a rain shadow and the total annual precipitation was 304 mm for the
period 1961-1990 (Alexandersson et al., 1991). However the total annual precipitation has increased
since then and is now around 350 mm (Abisko Station meteorological data; www.polar.se/abisko).

The vegetation cover in the Abisko area ranges from remnants of boreal pine forest, through the
subalpine zone dominated by mountain birch forest, through the low alpine belt, which extends
from the treeline up to where *Vaccinium myrtillus* no longer persist, to the high alpine belt with
non-vegetated surfaces (Carlsson et al., 1999; Lantmäteriet, 1997). The footprint of the eddy covari-
ance tower is charaterized by wet fen with no permafrost present, and vegetation dominated by tall
graminoids (Jammet et al., 2015, 2017).

According to Brown et al. (1998), the Abisko area lies within the zone of discontinuous per-
mafrost. However, with the observed permafrost degradation during the last decades (Åkerman and
Johansson, 2008; Johansson et al., 2011) the area is now more characteristic of the "sporadic per-
mafrost" zone. Permafrost is widespread in the mountains (Ridefelt et al., 2008), but at lower eleva-
tions permafrost is only found in peat mires (Johansson et al., 2006).

Data from three sites from the Torneträsk catchment (within an area of 10 km) have been used for
this study. The principal sites are Storflaket and Stordalen peat mires. The active layer measurements
and the ground temperatures are monitored at the Storflaket site (Åkerman and Johansson, 2008;
Johansson et al., 2011) and the carbon monitoring, including the eddy covariance measurement, is
carried out at the Stordalen site. These two mire sites are very similar in terms of climate, soil profile
and permafrost characteristics. For comparison, additional soil temperature data is included from a
mineral soil site at the Abisko Scientific Research Station, which is not underlain by permafrost.





### 3.2 Bayelva (Svalbard)

The study site is located in the high Arctic Bayelva River catchment area, close to Ny-Ålesund on Spitsbergen Island in the Svalbard archipelago. The catchment area lies between two mountains, with
the glacial Bayelva River originating from the Brøggerbreen glacier. The West Spitsbergen Ocean Current warms this area to an average air temperature of about −13°C in January and +5°C in July; it also provides about 400 mm of precipitation annually, which falls mostly as snow. The area has experienced a significant warming since the 1960s related to atmospheric circulation patterns and in later years the lack of sea ice during winter (Hanssen-Bauer and Førland, 1998; Førland et al., 2012).
In bioclimatic terms the area represents a semi-desert ecosystem (Uchida et al., 2009).

The study site is located on Leirhaugen hill (25 m a.s.l.), on permafrost patterned ground mainly consisting of non-sorted soil circles or mud boils. The ground is mostly bedrock but is partly covered by a mixture of sediments, comprising glacial till and finer glacio-fluvial sediments and clays. The mud boils have bare soil centers (about 1m diameter) and a surrounding rim of vegetation including
low vascular plants (mainly grass, sedge, catchfly, saxifrage and willow), mosses and lichens (Ohtsuka et al., 2006; Uchida et al., 2006). The soils are mineral (described as 'silty loam') with low organic content, although there can be locally high concentrations of organic carbon, for example at the base of the soil profile (Boike et al., 2008a).

The area is characterized by maritime continuous permafrost with temperatures around -2 to -
3°C. The active layer thickness in general exceeds 1m and can reach as deep as 2m in some areas (Westermann et al., 2010). Recent recent climatic warming has become manifest in the permafrost temperatures (Christiansen et al., 2010).

The eddy covariance measurements were conducted on Leirhaugen hill (78°55.0'N, 11°57.0"E). Additional meteorological observations and ground temperature measurements are continuously
conducted at the Bayelva soil and climate monitoring station (Boike et al., 2003, 2008a; Roth and Boike, 2001) 100m away. Over the past decade the Bayelva catchment has been the focus of intensive investigations on soil and permafrost conditions (Roth and Boike, 2001; Boike et al., 2008a; Westermann et al., 2010, 2011), and the surface energy balance (Boike et al., 2003; Westermann et al., 2009). Details of the measurements are provided in Westermann et al. (2009); Lüers et al.
200 (2014).

### 3.3 Kytalyk

The Kytalyk site (70°50' N, 147°30' E, 10 m a.s.l.) is located in the Kytalyk reserve, 28 km northwest of the village of Chokurdakh in the Republic of Sakha (Yakutia), Russian Federation. The site is located between the East Siberian Sea (150 km to the North) and the transition zone between taiga
and tundra. Based on the data from Chokurdakh airport, the monthly mean air temperatures range between -34.2 °C (January) and +10.4 °C (July). There is a current tendency to warming in partic-





ular in autumn (Parmentier et al., 2011). Annual mean precipitation amounts to 232 mm, of which about half falls as snow.

Three major topographic levels occur around the measurement site. The highest level in the area
is underlain by 'Ice complex deposits' or 'Yedoma': ice-rich silt deposits (Schirrmeister et al., 2002; Gavrilov et al., 2003; Zimov et al., 2006). The measurement site is located on the bottom of a drained former thermokarst lake, and the site is bordered by the edge of the present river floodplain. Both on the floodplain and the lake bottom a network of ice wedge polygons occurs, in general of the low-centered type. The ice wedge polygons on the lake bottom have broad ridges that may coalesce into
low palsa-like plateas. In between these plateaus a network of diffuse, strongly vegetated drainage channels have developed., This network of plateaus and drainage channels locally masks the original polygon structure. The mosaic of low plateaus and ridges is dominated by *Betula nana*, the diffuse drainage channels are covered with a meadow-like vegetation of *Eriophorum angustifolium* and *Carex* sp., hummocky *Sphagnum* with low *Salix* dwarf shrubs, polygon ponds are covered with
mosses and *Comarum palustre*, deeper ponds where ice wedges have thawed, and drier areas are covered with *Eriophorum vaginatum* tussocks. The soils generally have a 10-40 cm organic top layer overlying silt. In case of wet sites, the organic layer consists of loose peaty material, composed either of sedge roots or *Sphagnum* peat, depending on the vegetation. Drier sites tend to have a thinner, more compact organic layer.

The area is underlain by continuous permafrost. The active layer ranges from ∼25 cm in dry, peat-covered locations to ∼50 cm in wet locations. On the floodplain the active layer may be locally thicker.

The eddy covariance tower is located at a distance of ca. 200 m from the station buildings (van der Molen et al., 2007). The tower footprint covers a wet northwestern and southeastern sector domi-
nated by *Sphagnum* and ponds, while the northeastern and southwestern sectors have drier vegetation types.

### 3.4  Samoylov

The Lena River Delta in northern Yakutia is one of the largest deltas in the Arctic. Samoylov Island (72°22'N, 126°28'E) lies within one of the main river channels in the southern part of the delta and
is relatively young, with an age of between 4 and 2 ka BP (Schwamborn et al., 2002). The annual mean air temperature on Samoylov Island from 1998–2011 was −12.5°C, with the coldest monthly temperatures (January and February) around −30°C, and maximum monthly temperature around 10°C (July and August) (Boike et al., 2013). The landscape on Samoylov Island, and in the delta as a whole, has generally been shaped by water through erosion and sedimentation (Fedorova et al.,
2015), and by thermokarst processes (Morgenstern et al., 2013). The proportion of the total land surface of the delta covered by surface water can amount to more than 25% (Muster et al., 2012).





The terrace where the study site is situated is covered in low-centred ice wedge polygons. In the depressed polygon centres, drainage is impeded due to the underlying permafrost, leading to water-saturated soils or small ponds. The mineral soil is generally sandy loam, underlain by silty river deposits, with a ∼30cm thick organic layer at the surface (Boike et al., 2013). The vegetation in the polygon centres and at the edge of ponds is dominated by sedges and mosses, and at the polygon rims, various mesophytic dwarf shrubs, forbs and mosses dominate (Kutzbach et al., 2007). The maximum summer leaf coverage of the vascular plants was estimated to be about 0.3, and the leaf coverage of mosses was estimated to be about 0.95 (Kutzbach et al., 2007). It is estimated that moss contributes around 40% to the total photosynthesis (Kutzbach et al., 2007).

Continuous cold permafrost (with a mean annual temperature of -10°C at 10 m depth) under-lies the study area to between about 400 and 600 m below the surface. The active layer depth is generally less than 1m, and typical snow depth around 0.2-0.4 m (Boike et al., 2013). Since observations started in 2006, the permafrost at 10.7 m depth has warmed by > 1.5°C (Boike et al. (2013); http://gtnpdatabase.org/boreholes/view/53/).

Additional detailed information concerning the climate, permafrost, land cover, vegetation, and soil characteristics of these islands in the Lena River Delta can be found in Boike et al. (2013) and Morgenstern et al. (2013). Analysis of the energy balance for the site is found in (Boike et al., 2008b; Langer et al., 2011a, b).

### 3.5 Zackenberg

The Zackenberg study site is located near the Zackenberg Research Station within the Northeast Greenland National Park (74°28'N; 20°33'W). High mountains (> 1000 m a.s.l.) surround the Za-ckenberg valley to the west, east and north, while in the south a fjord forms its boundary. The area has been covered by the Greenland Ice sheet several times. The climate is high Arctic with an annual mean air temperature of -9.0°C (1996-2014) and only June, July, August and September have mean monthly temperatures above 0°C. The annual mean temperature has increased by 0.06°C per year since 1996 with most rapid warming occurring during summer months (Abermann et al., 2017). The mean annual precipitation is 211 mm (1996-2014) of which most falls as snow; the water availability is thus regulated by topography and snow distribution patterns. The seasonal snow cover is charac-terized by large interannual variability with maximum snow depths ranging from 0.13 m in 2013 to 1.33 m in 2002 (Pedersen et al., 2016).

Most vegetated surfaces in the Zackenberg valley are located below 300 m.a.s.l., where the low-land is dominated by non-calcareous sandy fluvial sediments (Elberling et al., 2008). Mineral soil types dominate while peat soils have limited spatial coverage (Palmtag et al., 2015). At least five main plant community types can be identified: fens occurring in water-saturated areas (Dupontia psilosantha, Eriophorum scheuchzeri), grasslands in semi-sloping, wet-to-moist terrain (Arctagrostis latifolia, Eriophorum triste), Salix arctica snow-beds mostly in slopes with prolonged snow cover,



*Cassiope tetragona* heaths in drier, level ground in the central valley, and Dryas heath in dry and wind-exposed areas (Elberling et al., 2008). The study site is located within a *C. tetragona* tundra

heath, dominated by *C. tetragona*, *Dryas integrifolia* and *Vaccinium uliginosum*, accompanied by patches of mosses.

Zackenberg is situated within the continuous permafrost zone, and the landscape development is dominated by periglacial processes. Only the upper 45-80 cm of the soil (active layer thickness) thaws every summer. However, in a CALM (Circumpolar Active Layer Monitoring Network) field

close to the study site, the maximum thaw depth has increased with 1.0-1.5 cm per year since 1997 (Lund et al., 2014).

Several studies on soil and permafrost (Palmtag et al., 2015; Westermann et al., 2015), surface energy balance (Lund et al., 2014; Stiegler et al., 2016; Lund et al., 2017) and carbon exchange (Mastepanov et al., 2008; Lund et al., 2012; Elberling et al., 2013) have been published based on

data from this site. A rich data set is available from this site through the extensive, cross-disciplinary Greenland Ecosystem Monitoring (GEM) programme (www.g-e-m.dk).

## 4 Methods

### 4.1 Evaluation data

#### 4.1.1 Carbon dioxide flux

Eddy covariance half hourly $CO_2$ flux data and related meteorological variables used in this study are archived in the PAGE21 fluxes database (http://www.europe-fluxdata.eu/page21) which is part of the European Flux Database Cluster.

Flux post-processing was performed consistently for all the sites following the protocol applied for the Fluxnet 2015 data release (http://fluxnet.fluxdata.org/data/fluxnet2015-dataset), with customized

choices of the processing options. The applied scheme included: (i) a quality assessment/quality control procedure over single variables aimed at detecting implausible values or incorrect time stamps (e.g. by comparing patterns of potential and observed downward shortwave radiation at a given location); (ii) the computation of net ecosystem exchange (NEE) by adding the $CO_2$ flux storage term calculated from a single $CO_2$ concentration measurement point (at the top of the flux tower) and

assuming a vertically uniform concentration field; (iii) the de-spiking of NEE based on Papale et al. (2006) using a threshold value (z=5); (iv) NEE filtering according to an ensemble of friction velocity (u*) thresholds obtained by bootstrapping following the methods of Barr et al. (2013) and Papale et al. (2006) and selection of a u* threshold, different for each year, based on the highest model efficiency (Nash-Sutcliffe); (vi) the gap-filling of NEE time series with the marginal distribution

sampling (MDS) method (Reichstein et al., 2005).



Finally, NEE was partitioned into the gross primary productivity (GPP) and ecosystem respiration (Reco) components using a semi-empirical model based on hyperbolic light response curve fitted to daytime NEE data (Lasslop et al., 2010). The years of data available for each site are given in supplementary Table S1.

### 4.1.2   Soil carbon profiles

Typical soil profiles with data on soil organic carbon content were generated for each site. Based on extensive field campaigns in each study area, individual pedons for representative landscape and soil types were combined and harmonized. In brief, soils were classified and sampled from open soil pits dug down to the permafrost. Permafrost samples were collected through manual coring into the

permafrost at the bottom of the soil pit. In most cases, soils were sampled to a depth of 1 m. The harmonized soil profiles were generated by averaging several soil pedons per landscape type at a 1 cm depth resolution. For more detailed descriptions of field sampling and labortatory procedures see Palmtag et al. (2015); Siewert et al. (2015, 2016). Top 1m total soil carbon values were calculated from a weighted average of different typical profiles, based on the fractional coverage of landscape

types in the footprint area of the flux towers.

### 4.1.3   Snow depth

Snow depth was recorded using automatic sensors (except Abisko where it is manual). Snow depth from the Abisko mire (Storflaket) was recorded manually monthly (Johansson et al., 2013). Snow height at Samoylov and Bayelva was recorded hourly, and for Zackenberg 3-hourly (using sonic

range and laser sensors). Snow depth at Kytayk was measured by means of a 70 cm vertical profile made of thermistors spaced every 5 cm (2.5 cm between 0 and 10 cm height from the ground ). Data were logged every 2 hours and the snow-air interface level was identified by analyzing the profile patterns with a Matlab® routine calibrated to search for deviations between consecutive resistance readings above a given threshold. Years used for each site are given in supplementary Table S1.

### 4.1.4   Soil temperature

For Samoylov, Bayelva, Kytalyk and Zackenberg, soil temperature was recorded hourly using thermistors (Kytalyk set-up described in van der Molen et al. (2007)). Ground temperatures for Abisko mire were recorded at the Storflaket mire, at boreholes cased with plastic tubes and instrumented with Hobo loggers U12 (Industry, 4 channels) together with Hobo soil temperature sensors (Johans-

son et al., 2011). Years used for each site are given in supplementary Table S1.

### 4.1.5   Soil moisture

Continuous soil moisture measurements are only available for Bayelva, Samoylov and Zackenberg. At Samoylov and Bayelva, hourly volumetric soil water content was recorded (using Time Do-




main Reflectometry). At Zackenberg soil moisture was measured using permanently installed ML2x
Thetaprobes (Lund et al., 2014). Years used for each site are given in supplementary Table S1. In-
dicative soil moisture levels for Abisko mire were collected from May to October 2015 (Pedersen
et al., 2017), measured manually as volumetric soil water content integrated over 0-6 cm depth using
a handheld ML2x Theta Probe (Delta-T Devices Ltd., Cambridge, UK). Soil moisture was measured
5 times in each plot and averages were subsequently used.

### 4.1.6 Active layer depth


Active layer depth was measured at CALM grids at most of the sites. At Bayelva there is no CALM
grid, so active layer was estimated from soil temperature measurements and is given as an 'indica-
tive' value. Active layer thickness monitoring is determined by mechanical probing. A 1 cm diameter
graduated steel rod is inserted into the soil to the depth of resistance to determine the active layer
thickness (Åkerman and Johansson, 2008) according to the CALM standard.

### 4.1.7 Leaf area index

Leaf area index was taken from MODIS product (MODIS15A2), for the closest coordinates to the
sites. This product has been successfully applied to tundra sites (Cristóbal et al., 2017). It was eval-
uated by Cohen et al. (2006) who found an RMSE of 0.28 at a tundra site. There are, however, still
considerable uncertainties in using this data product (see Section 5.6.1).

### 4.1.8 GPP per unit leaf area

This was calculated using the partitioned GPP from the eddy covariance data (Section 4.1.1), aver-
aged daily and taken on the same day as the values from the MODIS LAI product (Section 4.1.7).
Note that there are no time-resolved GPP values for Bayelva due to insufficient data. The extracted
GPP values were divided by the appropriate LAI estimates and the resulting values were collected
for all sites and binned into intervals of air temperature ($1.5°C$) and shortwave radiation ($20\,\mathrm{Wm}^{-2}$),
for which the mean and standard deviation were then calculated (shown on Figure 9).

### 4.2 Simulation set-up

The sites were represented in all the models by a single vertical column, although there was some
horizontal representation by means of tiling approaches (see model descriptions, Section 2). The
models were run in the most 'up-to-date' configurations, including new permafrost-relevant model
developments where available. Variables were output at hourly and/or daily resolutions.

  The meteorological driving data were prepared using observations from the site combined with
reanalysis data for the grid cell containing the site. For the period 1901-1979, Water and Global
Change forcing data (WFD) was used (Weedon et al., 2011). Data is provided at half-degree reso-
lution for the whole globe at 3-hourly time resolution from 1902-2001. For the period 1979-2014,



WATCH Forcing Data Era-Interim (WFDEI) was used (Weedon, 2013). For the time periods where observed data were available, correction factors were generated by calculating monthly biases relative to the WFDEI data. These corrections were then applied to the time-series from 1979-2014 of

the WFDEI data. The WFD before 1979 was then corrected to match this data and the two datasets were joined at 1979 to provide gap-free 3-hourly forcing from 1901-2014. Local meteorological station observations were used for all variables except snowfall, which was estimated from the observed snow depth by treating increases in snow depth as snowfall events with an assumed snow density (see Appendix). These reconstructions were then used to provide correction factors to WFDEI and WFD.

This leads to a more realistic snow depth in the model than using direct precipitation measurements, due to wind effects and the difficulty of accurately measuring snowfall. For Abisko, meteorological data from the research station were used, but additionally corrected by scaling the snowfall according to the ratio of monthly snow depths at the mire vs the research station (snow depth was only measured monthly at Storflaket mire), and a reduction of 1°C in air temperature.

Spin-up was performed as consistently as possible between the models, using the meteorological forcing from 1901-1930. Years were selected at random from this 30 year period and the models were run for 10000 years with pre-industrial $CO_2$ (1850, 286 ppm), followed by 50 years with changing $CO_2$ (1851-1900). The model state at the end of this spin-up period was taken as the initial state for the main run (1901-01-01 to 2013-12-31). For JSBACH, there was an initial 50 years of hydrological

spin-up before the main spin-up, with the permafrost impact on hydrology switched off, to allow the water to form a realistic profile (permafrost layers are impermeable and thus unrealistic initial conditions could otherwise be preserved). For JSBACH, the long spin-up was also between 7000-8000 years rather than 10000, since in this model there is no vertical representation of soil carbon, and therefore the soil carbon pools equilibrate much more quickly and had reached a steady state

after 7-8000 years. The $CO_2$ forcing data is from Meinshausen et al. (2011).

The soil parameters in the models were set up to represent each site as closely as possible (see Appendix, and Table A.1). These drew from literature values, a PAGE21 deliverable 'Catalogue of physical parameters', and field experience. (Note that the soil carbon profiles described in Section 4.1.2 were not used for this).

Vegetation was prescribed in ORCHIDEE and JSBACH. Since these are tundra sites, JSBACH used a 'tundra' PFT (100% coverage), which is similar to C3 grass but with reduced Vcmax (maximum rate of carboxylation in leaves). ORCHIDEE prescribed C3 grass (100% coverage) as there is no tundra PFT in this model version. JULES was run with dynamic vegetation using 9 PFT's (Harper et al., 2016), which do not include any tundra PFT's. All 9 PFT's prognostically determine

their coverage according to the environmental conditions, and they are all allowed to compete for space. In practice, only the C3 grass PFT is able to grow at these sites.

Some experiments were performed to separate the impacts of different processes. ORCHIDEE was run with and without vertical mixing of soil carbon. JSBACH carbon fluxes were analysed with



and without an additional contribution from a new moss photosynthesis scheme. In JULES, an extra
set of simulations was performed with fixed vegetation, to compare with the dynamic vegetation
scheme.

## 5    Results and discussion

The carbon dynamics are intrinsically linked to the physical state of the system, so we start by as-
sessing the snowpack, soil temperature, soil moisture, and active layer thickness in all three models.
The model physics has also been evaluated in detail in previous publications (Ekici et al., 2015,
2014; Chadburn et al., 2015a; Porada et al., 2016), so is kept short here. We then evaluate the soil
carbon stocks and the ecosystem $CO_2$ fluxes, and we analyse the $CO_2$ fluxes in detail. The fluxes
depend on every part of the system, so all of the preceding analysis contributes to our understanding
of the carbon dynamics at these sites.

### 5.1    Snow

Seasonal cycle of snow depth is shown in Figure 1. It depends strongly on the snowfall driving data.
Since the snowfall was back-calculated from the snow depth, the accumulation period should match
well with observations. There is still some variation due to the fresh snow density in the models
(which can differ both from the assumed density in making the driving data, and between the mod-
els), and furthermore the compaction of the snow is dependent on the model process representation
and physical conditions. Nonetheless, the models all make a reasonable simulation of the snowpack
accumulation and compaction. However, during the melting season they are less accurate, with the
snow often melting a little too early. Our method of back-calculating snowfall from snow depth may
miss some snowfall events during the melt season. There are also many other potential influences
such as albedo effects, snow-vegetation interactions and the influence of wind-blown sediment. For
example, the vegetation in the models is quite tall (up to 1m), and can lead to a lower albedo in the
models than reality, and thus faster snowmelt. At Bayelva, where the vegetation is particularly small
(∼5cm), there is a notable underestimation of the snow depth and early snowmelt in all models,
which supports this hypothesis (snow at Bayelva can be modelled very well when vegetation is not
included (López-Moreno et al., 2016)). Snowdrift is only represented by scaling the snowfall data to
match the observed snow accumulation, which limits the extent to which snowpack dynamics can
be recreated by the models.

### 5.2    Soil temperature

Soil temperature annual cycles at ∼40cm depth are shown on Figure 2. In general the models simu-
late the soil temperature at mineral soil sites quite well: Bayelva and Zackenberg sites on Figure 2.



There are greater errors in the simulation of organic soils: Abisko, Kytalyk and Samoylov on Figure 2.

For JSBACH and ORCHIDEE, the annual cycles of temperature are too large for the organic sites, indicating that these models need to better represent the insulating/damping properties of organic soils. To illustrate this, additional observations are shown on the Abisko plot (Fig. 2), from mineral soil at the nearby research station (where there is no permafrost). This line matches much more closely with the ORCHIDEE and JSBACH simulations, suggesting that these models are behaving thermally like a mineral soil. At Abisko, permafrost only occurs in peat plateaus and thus including organic soil properties in the models is essential for capturing the difference between permafrost and non-permafrost conditions.

In JULES, on the other hand, the annual cycle amplitude is too small at the organic sites and also at Zackenberg, mostly due to biases in the winter soil temperatures. This suggests that the snow thermal conductivity or density may be too low in JULES. A similar problem was found with a previous JULES simulation of Samoylov island, using a similar model set-up and forcing data (Chadburn et al., 2015a). There, the winter soil temperature was improved by increasing snow density.

### 5.3 Soil moisture

As with temperature, the (unfrozen) soil moisture is simulated well at mineral soil sites - see Bayelva and Zackenberg in Figure 3. In the winter, ORCHIDEE has a problem in that it does not represent the unfrozen water fraction in frozen soils, but the other models simulate a reasonable water content in winter. However, soil moisture is in general too low at organic sites - Samoylov and Abisko mire. The soils should be able to hold water near the surface and remain saturated very close to the surface (or even above). This points to problems with the hydrology schemes. The soil moisture is very important for the soil temperatures, and it can also have a strong influence on soil carbon stocks and the partitioning of decomposition into $CO_2$ and methane. Furthermore, it is important for moss photosynthesis, and therefore the uptake of $CO_2$ from the atmosphere. Therefore it is important to further improve the soil hydrology in these models.

Note that saturated zones can be influenced by landscape heterogeneity and lateral water fluxes that would not be captured in a point simulation. This can potentially be simulated by the models as a landscape average (see for example Gedney and Cox (2003)). However, such schemes simulate only a gridbox mean water content, which does not capture, for example, the influence of anaerobic conditions on decomposition.

Figure 3 shows quite a large variation in the timing of freeze-up and thaw between the models, reflecting the soil temperature differences in Figure 2. Correspondingly, the largest differences are at the organic soil sites.





### 5.4 ALT

The active layer depth is shown on Figure 4. In the models it is calculated by interpolation of soil temperatures to find the daily thaw depth, except in JULES which uses the method of Chadburn et al. (2015a). (The two methods differ at most within the thickness of the soil layers, Table A.2). In ORCHIDEE and JSBACH the active layer is too deep, which corresponds to the too-warm soil temperatures in summer, Fig. 2. In JSBACH the summer temperatures are only a little warmer than the observations - certainly closer than in ORCHIDEE, yet at some sites the active layer is just as deep. This is because technically the ALT cannot be diagnosed correctly in JSBACH, given the thick soil layers below 20 cm depth (see Appendix Table A.2). Increasing the resolution of the soil layers, while it does not make a big difference to the soil temperature profile, has a very large impact on the simulation of the active layer depth, as shown by Chadburn et al. (2015b). In JULES there is generally quite a good match to the observations as supported by the fact that the summer soil temperatures match closely with the observations for most sites. For Zackenberg the active layer is a little too shallow, but still in the range of observed values. This shows the importance both of resolving the soil column and the insulating effects of organic matter for determining the summer soil temperatures (Dyrness, 1982).

### 5.5 Soil carbon stocks

JULES and ORCHIDEE represent a vertical profile of soil carbon, whereas JSBACH does not. Without a vertical representation of soil carbon it is not possible to simulate permafrost carbon stocks, because all of the carbon is subject to the seasonal freezing and thawing of the active layer and the model does not contain any 'inert' permanently frozen carbon. Therefore, a vertical representation of soil carbon is prerequisite for simulating soil carbon stocks at these sites. However, JULES and ORCHIDEE have some problems in simulating the profiles - Figure 5. The biggest problem is underestimation: there is very little carbon simulated at many of the sites. For the sites where the quantity of soil carbon is somewhat realistic, the shape of the profiles vary from a steep exponential-looking decay with depth, to a shallower decline with more carbon in the deeper soil. The same kind of profiles are seen in the observations, particularly for the mineral soil sites (Bayelva and Zackenberg). However, neither of the models can produce the carbon-rich peaty layers of the organic soils. To simulate this would require additional process representation in the models, including representing saturated (and thus anaerobic) conditions in peat soil, and a dynamic representation of bulk density.

The reasons for the major underestimation are different in JULES and ORCHIDEE. In JULES, the main problem is that the GPP is underestimated, so there are not enough plant inputs to accumulate carbon in the soil. This is made clearer by Figure 6, which shows the relationship between GPP and top 1m soil carbon stocks. In JULES, the relationships are very similar to the observations, which indicates that the turnover of carbon in the soil is reasonable in JULES. Therefore, if the GPP



were large enough, the soil carbon stocks would be much more realistic. In ORCHIDEE, the story
is different. Even when the vegetation is productive, the soil carbon stocks are still very low. This
indicates a problem with the soil carbon decomposition. There are two factors that could affect this.
Firstly, the soil temperatures in ORCHIDEE are much too warm, and the active layer is too deep
(Fig.s 2 and 4). This can lead to too much decomposition. In order to improve this the model needs
to better represent the insulation from the organic soils. Another possible problem is the deep soil
respiration. In ORCHIDEE the only factor that suppresses the soil respiration at depth is the cold
and/or frozen nature of the ground. In JULES, however, there is an additional decay of respiration
with depth that empirically represents some processes that are missing in the model (following the
implementation in CLM, see Koven et al. (2013)). Including this in ORCHIDEE could lead to a
higher carbon stock at depth. The deeper soil carbon stocks are also influenced by long-term burial
processes, which are only represented by a simple diffusion scheme in these models. We include
JSBACH on Figure 6 because the top 1m soil carbon is mostly in the active layer. However, given
that the decomposition in JSBACH is controlled by the temperature of the top soil layer (3cm), it is
not surprising that the model somewhat underestimates the carbon stocks.

It should be noted that the observed relationship on Figure 6 may be confounded by the history of
soil carbon formation at these sites. There is inconsistency between Holocene climate and the pre-
industrial climate used in model spin-ups. Reconstructed Holocene climate for northern hemisphere
is warmer than pre-industrial (Marcott et al., 2013), and possibly wetter, favouring the formation of
peat, so some underestimation by the models may be expected.

We conclude that improving soil carbon stocks demands a different priority in each model. For
JULES, the first priority is to simulate realistic vegetation productivity, for ORCHIDEE it is to
improve the soil carbon decomposition, and for JSBACH it is to represent a vertical profile of soil
carbon. Assuming we can combine the best features from all of the models, the greatest difference
between the observed and simulated profiles will be the peaty, organic layers that are present in
observations and not models (Figure 5). Therefore the next priority for model development is to
better represent these organic soils. See e.g. Frolking et al. (2010); Schuldt et al. (2013) for examples
of modelling peat. While peatlands represent a small fraction of the land surface, they contain very
large carbon stocks (Yu et al., 2010), so it is important to include them in ESM's.

### 5.6 Carbon fluxes

Figure 7 shows the seasonal cycle of $CO_2$ flux at every site. The day-time and night-time fluxes are
plotted separately (partitioned by incoming shortwave radiation), showing in general uptake during
the day and emissions during the night. For the most part the models show uptake and emissions
at the same time as the observations, and a similar timing of peak uptake/emission (one exception
being the spring daytime flux in ORCHIDEE, see Section 5.6.1).



From the observations we also have the gap-filled estimates of annual gross primary productivity (GPP) and ecosystem respiration (Reco), which are compared with the annual totals for each model on Figure 8 (the moss GPP shown here is discussed in Section 5.6.3). For the GPP we see that for each model there is a positive correlation (sites with larger GPP in reality have larger GPP in the models), but that the overall values are too small for JULES, for ORCHIDEE there is a bigger

variation, and for JSBACH, they tend to be too large for the less productive sites and too small for the more productive sites - i.e. the slope of the relationship between model and observations is too shallow. Nonetheless, a significant amount of the variation between sites is captured by the models, to which the only inputs are climate data and soil properties. Of these, climate is the main driver of vegetation growth in these models (the soil only impacts the vegetation through moisture stress -

which is also partly climate-related), so we can say that a lot of the difference between the GPP/Reco across different sites is due to the difference in climate. In fact, in JULES and JSBACH, over 90% of the variation in GPP between sites is explained by the model, despite the systematic biases (R squared values of modelled GPP against observed GPP: JSBACH - 0.94, JULES - 0.95, ORCHIDEE - 0.63). This suggests that a model based on climate alone and with one tundra PFT could capture

most of the variability in tundra carbon uptake, if the vegetation was correctly calibrated. This is a promising sign that the model simulations could be easily improved.

  Due to the magnitude of errors in GPP and Reco, when considering the difference between the two - the net ecosystem exchange (NEE), the noise will be larger than the signal. Nonetheless, the models and observations both generally show a carbon sink in the present day, due to environmental

conditions being more favourable for growth (warmer, more $CO_2$) than in the 'pre-industrial' spin-up period (Table 2).

### 5.6.1 Drivers of carbon fluxes

The models indicate different drivers of GPP in different parts of the growing season. In particular, that GPP depends mostly on LAI until around the middle of the growing season (end of July) and

mostly on shortwave radiation in the second half of the season (August onwards). There is also a temperature dependence in all parts of the growing season. These relationships are shown in Supplementary Figure S1. Figure S1 also shows the plant respiration in the models, which exhibits a similar behaviour to the GPP, being influenced by temperature, shortwave radiation and LAI. The fact that these variables influence the GPP and autotrophic respiration is clear from the model structure (for

example Knorr (2000); Clark et al. (2011)), however the apparent split between the two halves of the season is an emergent behaviour.

  The other component of the ecosystem respiration is heterotrophic respiration. This does not exhibit the same dependencies as the plant respiration as it is determined by below-ground conditions. The heterotrophic respiration has a loose relationship with air temperature and a much stronger re-

lationship with the ∼20cm soil temperature - see Supplementary Figure S2.



In order to compare the photosynthesis schemes in the models more directly, we normalise by the LAI. It then becomes clear that the photosynthesis models in JSBACH and ORCHIDEE are in fact quite similar. Figure 9 shows the normalised GPP (per m$^2$ of leaf) against the air temperature and shortwave radiation. JSBACH and ORCHIDEE show similar relationships, although ORCHIDEE

still has a slightly higher GPP, potentially explained by the fact that Vcmax is higher. On these plots we also show the limited data that we can plot from observations, using MODIS LAI. It is clear that the normalised GPP in JULES is too low (this is a problem requiring attention in the model, probably related to canopy scaling), but for JSBACH and ORCHIDEE the GPP is approximately consistent with the observations. The observations are a little higher than the models, but this is largely in-

fluenced by underestimated LAI at Samoylov (note that for the other sites, MODIS LAI compares reasonably with ground-based estimates). Moss cover is close to 100% on Samoylov (Kutzbach et al., 2007) and by contrast, maximum LAI from MODIS is only around 0.3. This could be due to the large size of the MODIS pixels (1km×1km) leading to the inclusion of water in the pixel, or because the moss has a different absorption spectrum from vascular plants and could register as

bare soil. Whatever the cause, the GPP per unit LAI at Samoylov would be at least doubled by this underestimation of LAI, and if we were to account for this, the observation-based estimates would be very close to the JSBACH and ORCHIDEE results.

Aside from the low-bias in JULES, we therefore conclude that the main source of error in the modelled seasonal cycle of GPP is the huge variation in the simulated LAI. This is shown on Figure

10. For example, ORCHIDEE LAI remains at zero in the early season, when the observations and other models show carbon uptake, and it suddenly increases to a very large value later in the season, then showing an uptake that is much larger than the observations (Fig. 7). In fact, at Zackenberg the cumulative temperature is never high enough to initiate budburst in the model, so the LAI is always zero. These problems lead to unrealistic daytime emissions during spring from ORCHIDEE on Fig.

7 for most sites, and no fluxes at all for Zackenberg. Since the GPP seems to be consistent with observations when the impact of LAI is removed, we conclude that if the models could simulate the correct LAI they would largely simulate the correct GPP. JULES captures more of the difference in LAI between the sites than the other models (and subsequently captures more of the inter-site variation in GPP). This is because JULES is running a dynamic vegetation scheme that allows the

vegetation fraction to vary. The LAI from JULES with fixed vegetation is also shown on Figure 10, and captures less of the inter-site variability. Therefore, both improving the LAI and including a dynamic vegetation scheme is the priority for improved simulations of tundra carbon uptake.

### 5.6.2 Components of respiration

If the system were in equilibrium, the annual mean ecosystem respiration would be equal to the

GPP. Thus, improving the simulation of GPP would by default improve the simulated respiration. However, the seasonal cycle of respiration is significantly different from that of GPP, due to the



heterotrophic component. (This is particularly true in cold climates as the soil temperature can lag a long way behind air temperature due to the latent heat of freezing/thawing.) Furthermore, the response of respiration to changing conditions must be correctly simulated, otherwise any shift from

the equilibrium state - a net source or sink of carbon - will not be correctly simulated.

It is difficult to compare the modelled respiration fluxes with the eddy covariance data (other than the annual mean). This is because the gases are assumed to be immediately emitted from the soil in the models, whereas in reality they can accumulate in the soil profile, and diffuse upwards with a significant delay. The accumulated gas may also be released from the soil in bursts, e.g. in the case

of Bayelva, where the bursts of emissions in the autumn season correspond to heavy rainfall events, which (it is hypothesised) may be forcing the gas out of the soil (J. Boike, personal communication). Similarly, strong autumn emissions of $CO_2$ from the soil were observed by chamber measurements at Zackenberg, due to the freezing of the active layer forcing out bubbles of gas (Mastepanov et al., 2013). Further difficulty is introduced since the heterotropic and autotrophic components cannot be

separated in the measurements. Therefore we cannot evaluate the soil respiration schemes in detail without direct measurements in the soil. However, one conclusion we can make is that for some models the soil carbon is approximately correct when the inputs to the system (GPP) are correct (Figure 6), which gives some indication that the decomposition models behave reasonably in these conditions.

### 5.6.3 Nutrient limitation and moss.

We have discussed the need for a dynamic vegetation model to capture the inter-site differences in LAI, as shown on Figure 10 where JULES using a dynamic vegetation model captures much more of the inter-site variability than the other models. However, looking more closely highlights some missing processes.

For example, the LAI at Bayelva is very small (close to zero) during the early part of the JULES simulation, but between around 2002-2006 it rapidly increases to around 1. To illustrate this transition, the fractional coverage of vegetation in JULES is shown on Supplementary Figure S3. In reality, vegetation cannot establish rapidly at a site such as this (even if climatic conditions become appropriate), because of the lack of a soil matrix and nutrients needed for plant growth, particularly nitrogen.

Vascular plants could take 100's of years to establish once climatic conditions become appropriate, due to the large timescales involved in soil development. The vegetation at Bayelva is mainly mosses and lichens, which can grow in nutrient-poor conditions, but photosynthesise more slowly than vascular plants (Yuan et al., 2014). Therefore, to simulate the $CO_2$ flux at a very nutrient-limited site it is necessary to have a different PFT that represents the low-nutrient but low-GPP vegetation such as

moss, and to include nutrient limitation for the other PFTs.

A similar problem can be seen at Samoylov, where around 90% of the site is covered by moss (Boike et al., 2013), and JULES simulates an LAI similar to that of Kytalyk (as the climatic con-





ditions are similar), but in reality the LAI's of the two sites are very different and at Samoylov the LAI (of vascular plants) and $CO_2$ flux should be much smaller than that of Kytalyk. At Samoylov, the moss contributes around 40% to the total photosynthesis (Kutzbach et al., 2007), showing its importance in the carbon budget of this site. It is hypothesised that there are fewer vascular plants at Samoylov because the more waterlogged conditions (due to many polygon centre ponds) could reduce vegetation growth. In fact, reduced vegetation growth is also seen in areas with many polygon centre ponds at Kytalyk. Moreover, nitrogen may be lost in these waterlogged environments by denitrification, making it a more nutrient-limited environment.

Thus, to really capture the inter-site differences in GPP it is necessary to include nutrient limitation and other soil/plant interactions in the model. And once nutrient limitation is introduced, then moss is required (which grows in nutrient-deficient and very wet conditions where the vascular plants will not grow) in order to recreate the observed carbon uptake.

In JSBACH, moss carbon fluxes can be included - see Figure 8. This shows that the moss model can contribute significantly to the carbon budget at the mossy sites. However, at the sites with less vascular vegetation in reality (Bayelva and Samoylov), including the moss makes the total fluxes much too large, as JSBACH (like JULES) simulates too much vascular vegetation.

At Samoylov there is an early-season peak of carbon uptake that is missed in the models (Figure 7). It is possible that this could correspond to the wet ground directly following snowmelt, which leads the moss to start photosynthesising. However, it is difficult to make conclusions from the data available, and we also know that eddy covariance methods can have some problems around the time of snowmelt (for example Pirk et al. (2016)). Nonetheless, we can get a clue from the moss model in JSBACH. Supplementary Figure S4 shows the annual cycle of moss GPP along with the GPP from JSBACH (without moss), showing that it captures an early-season peak before the vascular plant uptake starts in JSBACH. This plot also shows the moisture content of the moss layer, making it clear that there is a strong relationship between moisture content and moss photosynthesis. Thus it becomes even more important to simulate soil moisture correctly once moss is included in the models.

## 6 Conclusions

Based on the analysis above, we can identify priority developments that would improve the carbon stocks and fluxes in the models. Assuming that 'state-of-the-art' is represented by a combination of the best parts of each model, we provide the following priorities for next steps to advance the state-of-the-art:

1. Improve vegetation phenology/dynamics to simulate realistic LAI (including nutrient limitation and dynamic vegetation).

2. Include moss both for photosynthesis and peat accumulation.





3. Improve the soil carbon profile for organic soils (including peat processes).

There are feedbacks between the vegetation and the soil physical state (e.g. Sturm et al. (2001)),
so incorporating more realistic vegetation such as Arctic shrubs could also lead to an improved sim-
ulation of the soil temperature and moisture. There are several reasons why distinguishing between
different tundra PFT's, such as grasses and shrubs, could be useful, such as differences in carbon
storage, and snow interactions. Note that JULES includes a 'shrub' PFT, but these are large shrubs
($\sim$ 1.5m tall) which would not be expected to grow at the cold sites. Smaller, cold-tolerant shrubs
should be added as a separate PFT. There are few modelling studies to date where tundra phenology
is explicitly considered, but see Van Wijk et al. (2003) for one example.

In JSBACH the moss photosynthesis is already simulated, and the coupling to the soil carbon will
be available in the next version. This provides clear guidance for other models to follow, see Porada
et al. (2013, 2016). However, since JSBACH does not include nutrient limitation, the combined
GPP/Reco from vascular vegetation and moss is too high (Fig. 8). Including nutrient limitation is an
essential part of these priority developments.

In order to facilitate improvements to the vegetation schemes, better site-level measurements of
LAI are required. This was identified as one of the largest modelling uncertainties, but only indirect
satellite-derived LAI products are available, which are not sufficiently detailed or accurate for devel-
oping the model schemes. Furthermore, in order to improve the simulation of soil carbon profiles,
better observations and understanding of all below-ground processes such as in-situ decomposition
rates and the dynamics of cryoturbative mixing are required (Beer, 2016).

Future changes in NEE are key to understanding the role of the Arctic in a global context. We can
see in Table 2 that the size of the NEE is much smaller than the errors we are currently seeing in, for
example, the simulated GPP. This supports the need for the model improvements highlighted above.

Future changes in the carbon balance will come both from changes in vegetation productivity/type,
and decomposition of old soil carbon due to thawing permafrost. Therefore, dynamic vegetation (in-
cluding nutrient limitation) is required for future simulations as well as for simulating the correct
LAI in the present day. The vertical representation of soil carbon is therefore also particularly im-
portant for the fluxes in the future. However, soil carbon release will also be triggered by landscape
dynamics like ground collapse and thermokarst formation, which are not yet represented in any of
these models. See e.g. Schneider von Deimling et al. (2015) for a modelling study in which some of
these impacts are included. This is another important aspect that must be taken into account in future
model development (Rowland and Coon, 2015).

Accurate process representation at a site level will not necessarily transfer the same level of accu-
racy to a global simulation. In particular, there are issues with using a single 'gridbox mean' value
to represent a large area of land (heterogeneity in soil/microtopography exerts non-linear controls
on carbon and vegetation dynamics), and with obtaining realistic large-scale observations for quan-
tities such as soil parameters. On the other hand, the sites used in this study represent typical tundra



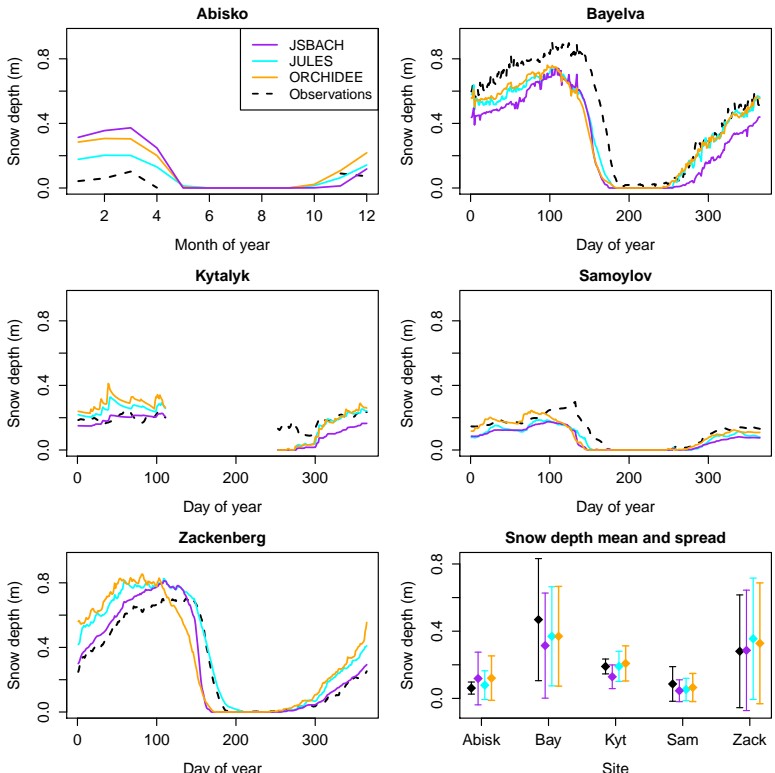

**Figure 1.** Mean annual cycle of snow depth at each site, showing both observations and models. See supplementary Table S1 for years used at each site.

sites, and the model development priorities that we identify are consistent across sites, indicating that these would also lead to improved tundra carbon dynamics in global simulations. This study has allowed us to quantify deficiencies in the models that we could not have robustly identified using global datasets, due to the quantity and quality of observational data available.

**Appendix A:  Details of model set-up**

Mineral soil properties were calculated from sand/silt/clay fractions. Slightly different pedotransfer functions are used in each model, but they are all taken from the same baseline soil texture (see Table A.1). For JULES, the organic soil fraction as a function of depth was estimated using the bulk density and carbon density. The combined organic/mineral soil properties were then calculated as in Chadburn et al. (2015a).




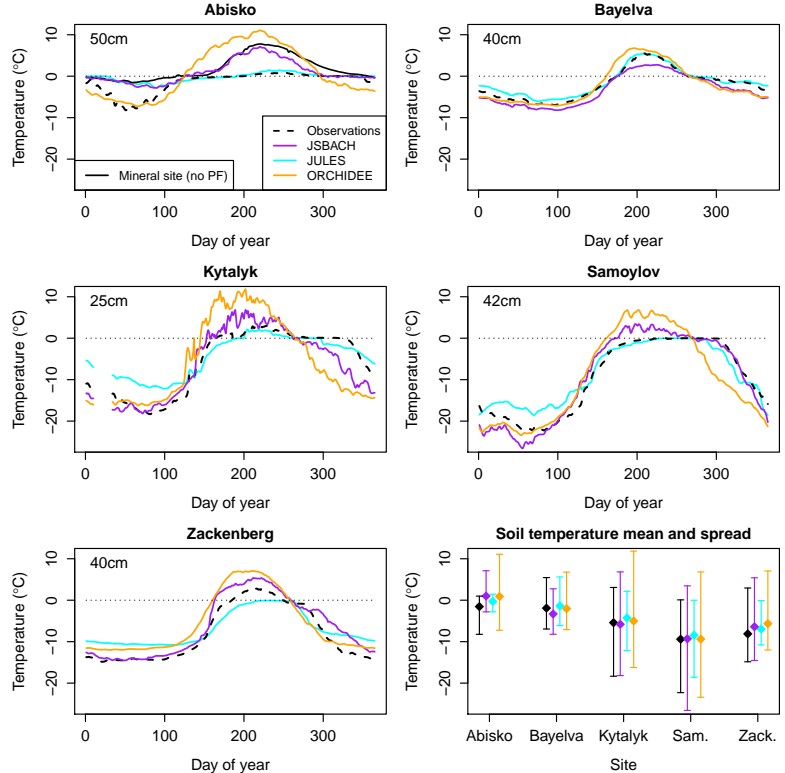

**Figure 2.** Mean annual cycle of soil temperature at each site, showing both observations and models. Depths of observations: Abisko: 50cm, Bayelva: 40cm, Kytalyk: 25cm, Samoylov: 42cm, Zackenberg: 40cm. JULES and ORCHIDEE take nearest soil layer and JSBACH is interpolated to correct depth, as soil layers are not well-enough resolved to get close to the right depth. See supplementary Table S1 for years used at each site.

**Table 1.** Key climatic/physical variables at the sites.

|  | Abisko | Bayelva | Kytalyk | Samoylov | Zackenberg |
|---|---|---|---|---|---|
| Mean annual air temp. | -0.6°C | -5°C | -10.5°C | -12.5°C | -9°C |
| Summer air temp. | 11°C | 5°C | 10°C | 10°C | 6.5°C |
| Winter air temp. | -11°C | -13°C | -34°C | -33°C | -20°C |
| Annual precipitation | 350 mm | 400 mm | 230 mm | ~190 mm | 210mm |
| Typical snow depth | 0.1m | 0.5-0.8m | 0.2-0.4m | 0.2-0.4m | 0.1-1.3m |
| Active layer depth | 0.55-1.2m | 1-2m | 0.25-0.5m | <1m | 0.45-0.8m |
| Permafrost temperature | ~0°C | -2 to -3°C | -8°C | -10°C | -6.5 to -7°C |
| Soil type (mineral/organic) | Organic | Mineral | Organic | Organic | Mineral |




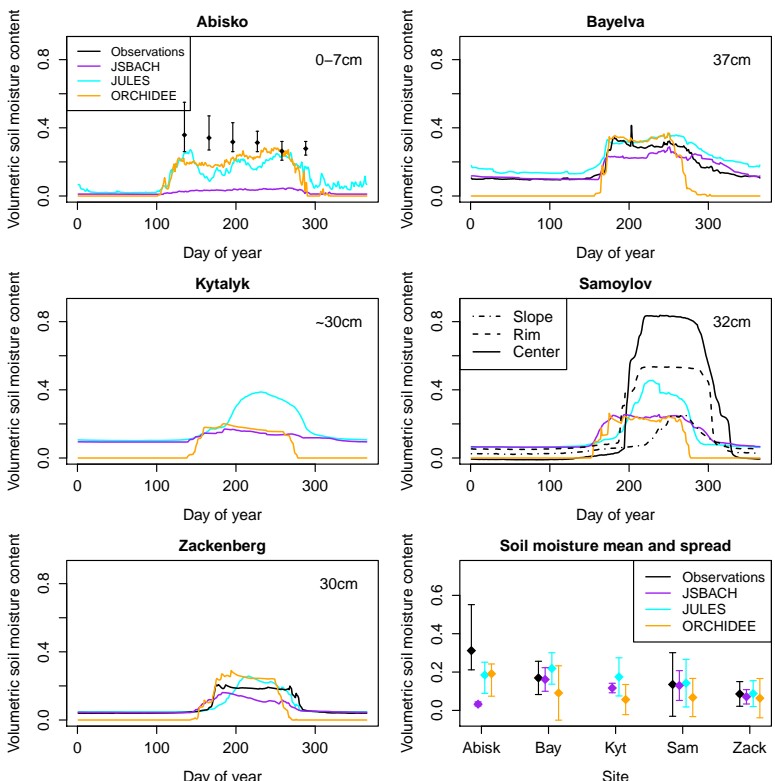

**Figure 3.** Mean annual cycle of unfrozen soil moisture at each site, showing both observations (where available) and models. Depths: JSBACH: 19cm for all sites (this is the closest to 30cm - the next layer is at 78cm), except Abisko, 3cm. JULES: 32cm (except Abisko, 3cm). ORCHIDEE: 36cm (except Abisko, 4cm). Observations: Bayelva: 37cm Samoylov: 32cm Zackenberg: 30cm Abisko: 0-7cm. For Samoylov, three different soil moisture profiles are shown that represent different parts of the polygonal microtopography. See supplementary Table S1 for years used at each site.

Assumed 'fresh' snow density for creating snowfall timeseries from snow depth: This depends on the resolution of the data. If we have low-resolution snow depth data, there may be some compaction between the snow landing and the measurement being taken, so we will use a higher density to generate the timeseries. The density used for most sites, hourly to daily resolution, is 180 $kgm^{-3}$. At Abisko, only 5-daily snow depth data was available, and this was at the research station rather

than the mire. Since this is a relatively warm site leading to more melting, and due to the long time interval between readings, in order to give enough snow in the models a density of 240 $kgm^{-3}$ was used. For Abisko mire there were just a handful of snow depth measurements each year. All available values taken during a given month were averaged to give a monthly average timeseries



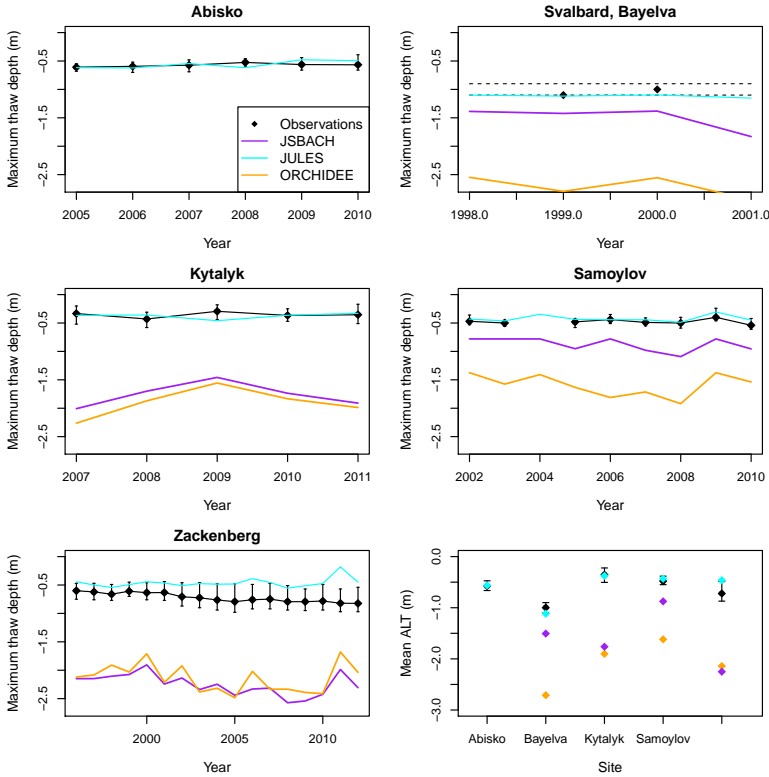

**Figure 4.** Maximum summer thaw depth (active layer) over a number of years at each site, comparing observations and models.

**Table 2.** Mean NEE budget ($gCm^{-2}yr^{-1}$), showing that in general this is smaller than the errors in simulated GPP, therefore the noise is larger than the signal in this data. Positive numbers represent a carbon source.

| Site | JSBACH | JULES | ORCHIDEE | Observations |
|---|---|---|---|---|
| Abisko | -6.6 | -16.0 | -79.2 | **-162.0** |
| Bayelva | -8.8 | -15.1 | -34.7 | **-13.9** |
| Kytalyk | -19.0 | -18.9 | -24.3 | **-108.0** |
| Samoylov | +1.5 | -15.1 | -58.9 | **-49.6** |
| Zackenberg | +35.9 | -5.2 | +0.01 | **-12.0** |
| *Mean absolute error in GPP* | *100.2* | *123.6* | *88.4* | *-* |

of snow depth. We compared the depth with the model output from JULES using the forcing data
prepared from the research station. The snowfall was then scaled according to the ratio of monthly





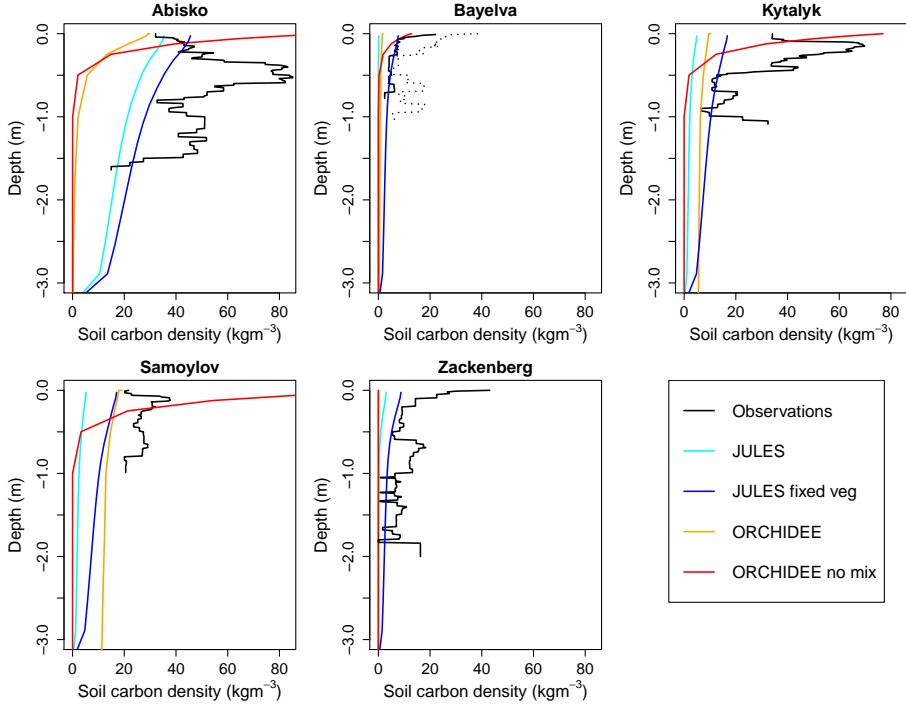

**Figure 5.** Profile of soil carbon at each site (kgm$^{-3}$). Observations and two of the models (ORCHIDEE and JULES) are shown, as these models have a vertically resolved soil carbon profile.

snow depth in the model vs the observations. This approach introduces uncertainties that would be reduced by the availability of a higher-resolution snow depth dataset from Stordalen mire.

*Acknowledgements.* The authors acknowledge financial support by the European Union Seventh Framework Programme (FP7/2007-2013) project PAGE21, under GA282700. SEC, SW and GH acknowledge support
from COUP (Constraining Uncertainties in Permafrost-climate Feedback) Joint Programming Initiative project (S.E.C: National Environment Research Council grant NE/M01990X/1; G.H: Swedish Research Council grant no. E0689701; S.W: Research Council of Norway project no. 244903/E10).





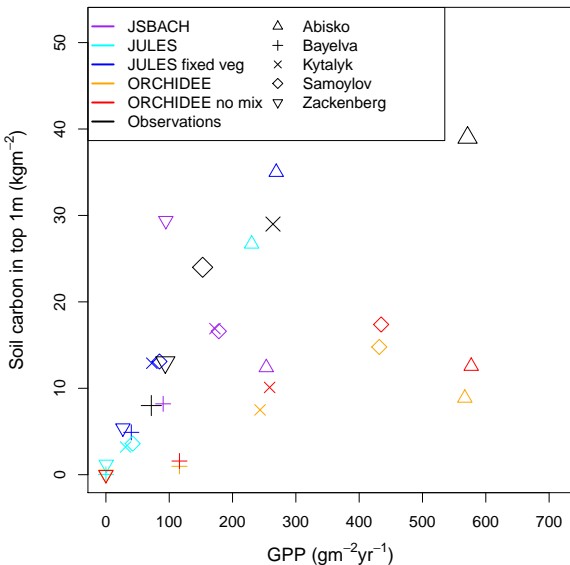

**Figure 6.** GPP against top 1m soil carbon at each site. The top 1m soil carbon values are for the tower footprint area (see supplementary Table S2), so that equivalent values are being compared.

**Table A.1.** Parameters for the sites. [1]Klaminder et al. (2008). [2]J. Boike, personal communication. [3]Van Huissteden et al. (2005). [4]van der Molen et al. (2007). [5]Boike et al. (2013). [6]Hollesen et al. (2011). [7]Rydén et al. (1980). [8]Roth and Boike (2001). [9]S. Zubrzycki, soil carbon data. [10]PAGE21 Catalogue of physical parameters. [11]Bartholomeus et al. (2012). [12]Elberling et al. (2008). In most cases soil types (see Section 3) were translated to approx. sand/silt/clay fractions using Table 1 in Beringer et al. (2001). *Estimated from bulk density. Topographic index is from a global dataset: a 0.5° aggregate from (US Geological Survey, 2000).

|  | Abisko | Bayelva | Kytalyk | Samoylov | Zackenberg |
|---|---|---|---|---|---|
| Latitude | 68.35 | 78.92 | 70.83 | 72.22 | 74.5 |
| Longitude | 19.05 | 11.93 | 147.5 | 126.28 | -20.6 |
| Organic layer thickness (cm) | ∼50 [1] | 0 [2] | ∼20 [3,4] | ∼30 [5] | 5 [6] |
| Sand fraction | 0.1 | 0.17 | 0.17 | 0.58 | 0.8 [6] |
| Silt fraction | 0.9 | 0.7 | 0.7 | 0.32 | 0.1 [6] |
| Clay fraction | 0.0 | 0.13 | 0.13 | 0.1 | 0.1 [6] |
| Bulk density | 1.3 [7] | 1.7 [8] | 0.6 [4] | 0.8 [9] | 0.9-1.8 [10] |
| C below organic layer ($kgm^{-3}$) | 14* | 0* | 17 [11]* | 35 [9] | 10 [12] |
| Topographic index mean | 4.0 | 3.9 | 6.2 | 5.9 | 6.7 |
| Topographic index st.dev. | 2.5 | 1.6 | 2.8 | 2.2 | 1.2 |

**Table A.2.** Soil layer thicknesses in the models.

| Model | Layer thicknesses (m) |
|---|---|
| JSBACH | 0.03, 0.19, 0.78, 2.68, 6.98, 16.44, 38.11 |
| JULES | 0.05, 0.08, 0.11, 0.14, 0.17, 0.19, 0.22, 0.24, 0.26, 0.28, 0.30, 0.32, 0.34, 0.36, 0.38, 0.40, 0.42, 0.44, 0.46, 0.47, 0.49, 0.51, 0.53, 0.54, 0.56, 0.58, 0.59, 0.61 |
| ORCHIDEE hydrological | 0.0005, 0.002, 0.01, 0.01, 0.03, 0.06, 0.12, 0.25, 0.50, 1.00, 1.75 |
| ORCHIDEE thermal | 0.0005, 0.002, 0.01, 0.01, 0.03, 0.06, 0.12, 0.25, 0.50, 1.00, 1.75, 2.50, 3.50, 4.55, 5.66, 6.81, 8.03, 9.31, 10.65, 12.06, 13.54, 15.09, 16.72, 18.43, 20.23, 22.12, 24.10, 26.18, 28.37, 30.66, 33.07, 35.60 |

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




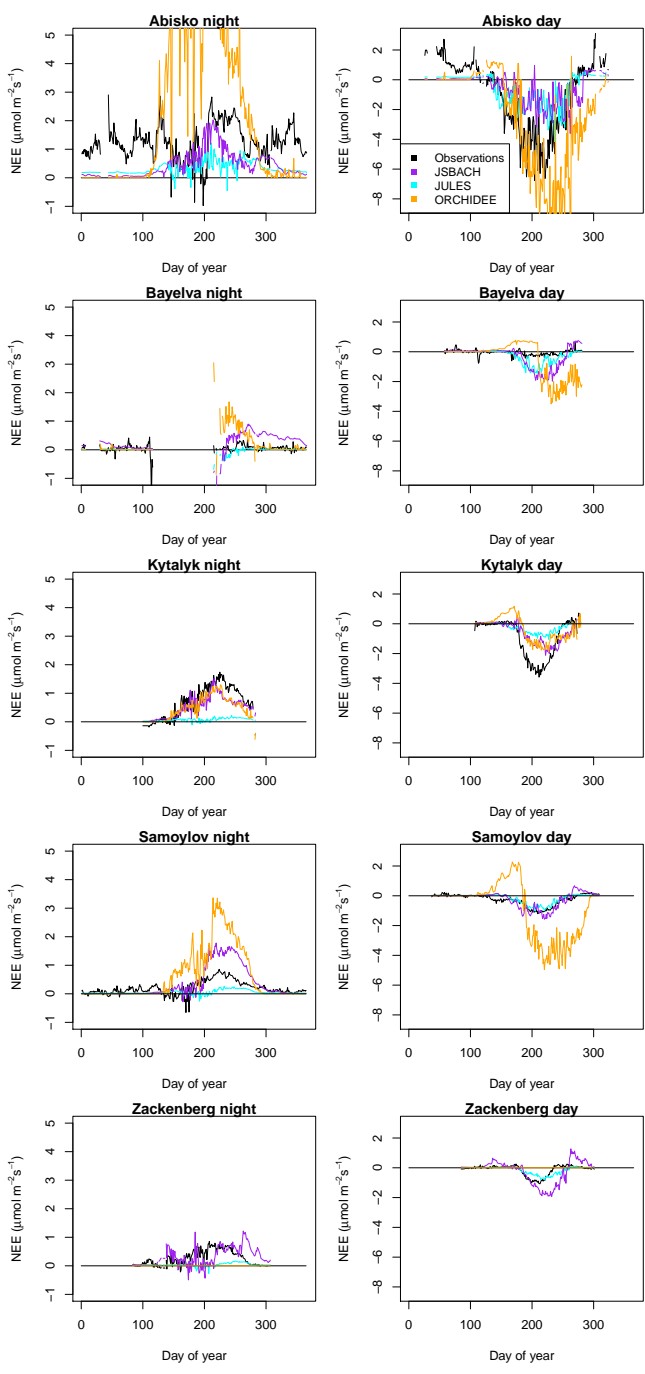

**Figure 7.** Mean annual cycles of $CO_2$ fluxes for all sites, observations and models. Left: nightime flux; Right: daytime flux (corresponding to incoming shortwave radiation $>20$ Wm$^{-2}$). See supplementary Table S1 for years used at each site.



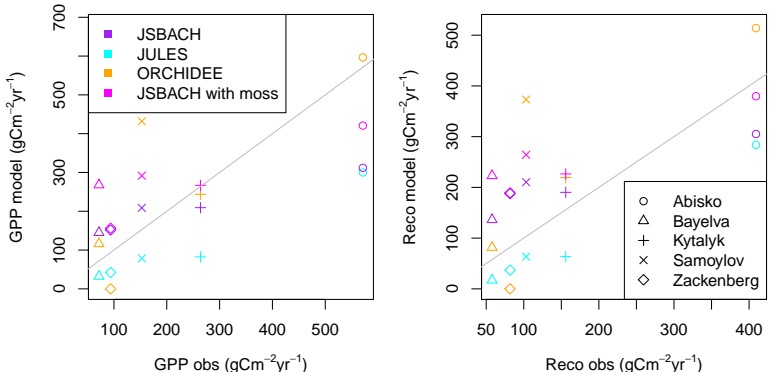

**Figure 8.** Mean annual GPP (gross primary productivity) and Reco (ecosystem respiration) from the models, plotted against the observation-derived values for the same time periods. See supplementary Table S1 for years used at each site.

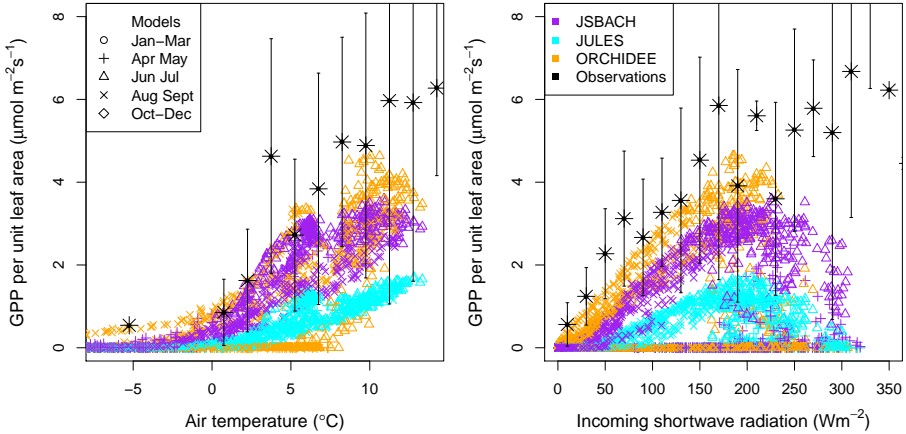

**Figure 9.** Relationship of 'normalised' GPP (GPP per m$^2$ of leaf) to air temperature and incoming solar radiation. All models and sites are shown, plus observationally-derived values using GPP estimated from eddy covariance data and LAI from MODIS (MODIS15A2), see Section 4.1.8.





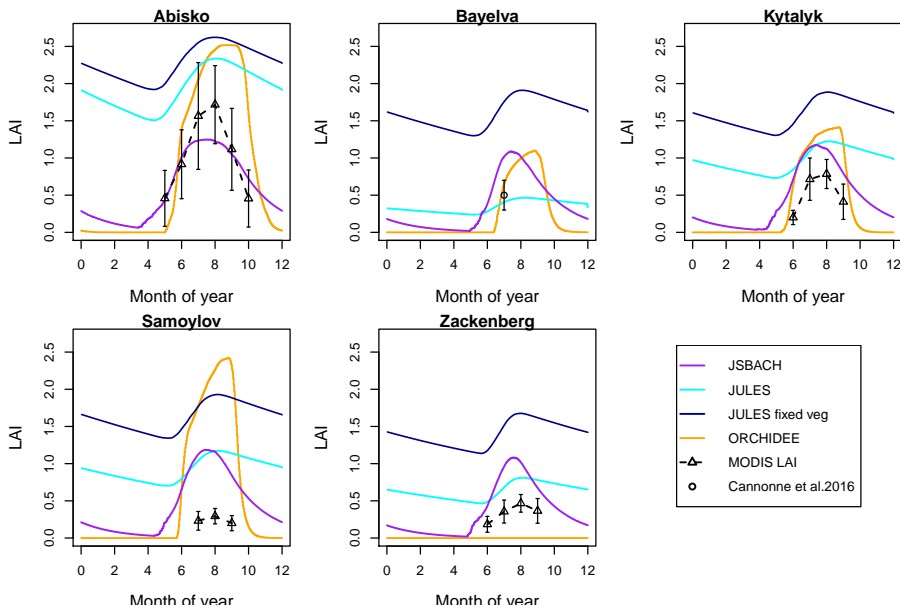

**Figure 10.** Mean annual cycles of LAI (leaf area index), for each site. 'Observed' values are from MODIS LAI product (MODIS15A2), except Bayelva which is from Cannone et al. (2016).