# Peer review of "Carbon stocks and fluxes in the high latitudes: Using site-level data to evaluate Earth system models"

_Biogeosciences, 2017_

## Referee Comment (RC1) · F. Dominé (Referee) · 29 Jun 2017

This paper evaluates the ability of 3 land surface schemes used in earth system models to simulate carbon stocks and fluxes at 5 Arctic sites. The exercise is timely and welcome, given that permafrost carbon feedbacks are not included in most climate projections. The breadth of activities dealt with in this paper is very large and includes Arctic field measurements and observations, model developments and model runs. Topics include physics, chemistry and biology, and this work needed the varied competences of tens of authors. An in-depth review of this paper by a single person therefore appears extremely difficult and I first need to state the limits of my review. Although I do

use land surface schemes to model Arctic permafrost and snow, I am mostly a field scientist focused on performing detailed and sometimes complex field measurements and analyzing them. I also focus on snow and thermal processes in snow and soil, as well as on vegetation-snow-soil interactions. My review should therefore be complemented by that of a modeling expert, preferably focused on carbon aspects. Overall, I found the paper very interesting in that it shows that the current state of the art in modeling Arctic carbon cycling is clearly below our needs for reliable projections and it identifies key aspects where progress is most needed. Given the form of the paper, performing additional simulations would probably be difficult, but I recommend minor but significant changes in text organization, in the use and critical evaluation of field data, and perhaps minor modifications to the conclusions.

**Model description**

How about a Table summing up the 3 models main features? This would allow significant text shortening. Also please make sure equivalent information is given for all 3 models. For example, vegetation details are lacking for JSBACH. By the way, PFT is defined nowhere and some institute abbreviations are not explained (IPSL, NCSDC). I let the editor decide whether that is necessary. Lichens are not mentioned in any model description, from which I assume that they are not considered. Yet, they can be very abundant at some Arctic sites, sometimes covering most of the ground. They have physical and biological properties very different from mosses, for example a much lower thermal conductivity and different hydrological properties which strongly impacts the ground thermal and hydrological regimes. Please consider specifically mentioning this omission. A couple of sentences or a line in the future model Table to describe the snow scheme would be nice (single layer, multilayer...). Please also specify here that nutrient aspects are not treated in any of the 3 models.

**Site description**

The description of all sites should really be homogenized and considerably shortened,

by at least 3 pages. What is in Table 1 need not be repeated in the text. Incomparable data are often given in the text. For example, some sites have mean annual temperature, others January and July, please be consistent. Also detail the snow fraction of precipitation in all cases: "most" is vague and not very useful. All plant Latin names must be in italics. By the way, line 219, what are the Salix? Richardsonii, arctica, other?

**Field data**

Measuring snow precipitation and snow depth in a reliable and representative manner is always a problem in the Arctic and the text does not convince me that this aspect was treated properly. Moreover, its impact may be understressed here since it conditions the permafrost thermal regime and therefore all carbon processes. How about details of the precipitation measurement, such as the presence of a wind shield around the gauge? I understand that precipitation measurements were not used, but since snow depth measurements are not convincing, as detailed below, perhaps analyzing precipitation data in more detail would be useful. Was there any attempt to correct measured snow precipitation as described in (Forland et al., 1996)? This can double estimates of precipitation amounts and considerably improve agreement with snow accumulation. Measuring snow depth in a representative manner is difficult. Certainly using one point measurement is inadequate. In particular, in low-centered polygons, variations are huge and at least 100 measurements are required for a representative value. Please detail the representativity of your snow depth measurements. In case the data are found to have limited representativity, this should be clearly stated and perhaps a sensitivity study would be useful (if it is still possible to perform it): what is the impact of snow amount on permafrost temperature and carbon cycling? Perhaps looking at data from reanalyses would also be helpful for an extra evaluation of precipitation and snow depth data. Are there any field measurements of snow density to validate model assumptions of this variable? By the way, snow temperature measurements at several heights can be very useful to evaluate the validity of snow schemes, and implementing those at the sites described here may be valuable for future work (Barrere et al., 2017).
Lines 327 and 329: please use "snow depth" throughout.

**Results and discussion**

Lines 431-432. The snow depth model output at Abisko is not "reasonable". It just does not seem to work there. Please consider representativity of field data and modify discussion.

Line 433. "snow often melting a little too early" in simulations. Ambiguous as written.

Lines 436-437. How do models account for vegetation effects on snow albedo?

Lines 457-458. How about thermal conductivity values obtained by JULES, and how do they compare with other models? Perhaps also compare with values obtained at a comparable high Arctic sites in low-centered polygons (Domine et al., 2016) if you think this supports your case. Note by the way that stratification of thermal conductivity can have an important effect, as suggested by (Barrere et al., 2017), so that one-layer snow models can give the correct mean thermal conductivity value while making a large error on atmosphere-ground heat fluxes. Incorrect snow thermal conductivity stratification can also lead to incorrect timing of ground freezing and thawing. Arctic snow often has a very low thermal conductivity layer at the base, which delays freezing and thawing. This process is missed if the snow scheme gives a high thermal conductivity to the basal snow layer.

Line 559. A word on nutrients here?

Line 574-575. "GPP depends mostly [. . .]on shortwave radiation in the second half of the season". How about moisture? For example, (Frost and Epstein, 2014) stated that "rates of shrub [. . .] expansion were not strongly correlated with temperature trends and were better correlated with mean annual precipitation".

**Conclusion**

The impact of mosses is stressed, but as mentioned above, I really think that lichens

can have a huge impact. I gather that they are not very important at the sites studied here, but on a pan-Arctic scale, this is probably different.

Since you are talking about landscape dynamics, you may talk about the impact of lakes and ponds caused by landscape dynamics such as thermokarst lakes formation. These lakes are often hotspots of GHG emissions. See e.g. (Bouchard et al., 2015) and references therein.

Figure1. What is the meaning of mean snow depth? Spatial mean? Temporal mean? Over what period? The Abisko graph does not seem to match the mean value.

Table 1: What is summer? What is winter? Permafrost T, at what depth?

**References**

Barrere, M., Domine, F., Decharme, B., Morin, S., Vionnet, V., and Lafaysse, M.: Evaluating the performance of coupled snow-soil models in SURFEXv8 to simulate the permafrost thermal regime at a high Arctic site, Geosci. Model Dev. Discuss., 2017, 1-38, 2017.

Bouchard, F., Laurion, I., Preskienis, V., Fortier, D., Xu, X., and Whiticar, M. J.: Modern to millennium-old greenhouse gases emitted from ponds and lakes of the Eastern Canadian Arctic (Bylot Island, Nunavut), Biogeosciences, 12, 7279-7298, 2015.

Domine, F., Barrere, M., and Sarrazin, D.: Seasonal evolution of the effective thermal conductivity of the snow and the soil in high Arctic herb tundra at Bylot Island, Canada, The Cryosphere, 10, 2573-2588, 2016.

Forland, E. J., Allerup, P., Dahlstrom, B., Elomaa, E., Jonsson, T., Madsen, H., Perafü, J., Rissanen, P., Vedin, H., and Vejen, F.: MANUAL FOR OPERATIONAL COR-RECTION OF NORDIC PRECIPITATION DATA, DET NORSKE METEOROLOGISKE INSTITUTT, Oslo. REPORT NR. 24/96, 66 pp., 1996.

Frost, G. V. and Epstein, H. E.: Tall shrub and tree expansion in Siberian tundra ecotones since the 1960s, Global Change Biology, 20, 1264-1277, 2014.

---

## Referee Comment (RC2) · Anonymous Referee #2 · 12 Jul 2017

General Comments

This paper describes a study in which 3 land surface models are assessed against site-level evaluation data on some of the key indicators of carbon dynamics in high-latitude terrestrial ecosystems. The analysis and comparison of models were conducted such that the study identified particular, problematic issues for these models in capturing carbon cycling and related mechanisms important in simulating arctic ecosystem processes. From these, the authors recommend three issues as priority areas for model improvement in representing high-latitude components of earth system models. The paper presents a set of analyses that were well-designed to evaluate key processes in

these models. The paper is well-written and organized in such a way to give the audience a straightforward and easily understandable read of the methods and results of the study. A strength of this paper is the authors' interpretation and synthesis of the results into three, clearly outlined recommendations for model improvement. While none of these are necessarily "game changers" in the way we think about the problem, the results of this study will be of interest and use to those members of the modeling community who are working to build or improve their simulation of high-latitude ecosystem processes. Some additional thoughts on the motivation, objectives and hypotheses or expectations of this study might help define the scope of inference for this paper. But, the authors do take good care not to overreach on the conclusions, and appropriately offer several caveats of the analysis and remaining uncertainties that are not addressed with this study.

Specific Comments

While the study is laid out well in the paper, I do wonder if some re-structuring and/or additional information could strengthen the emphasis on the key ecosystem processes being studied here, i.e. the 6 or so pieces serving as subsections of the results. More text on background and motivation could be added to the introduction for each, including more references for previous studies that have explored these issues. Perhaps it'd be useful in the introduction to add a sentence on each issue that frames them as questions or hypotheses as a preview of what the reader can expect to see in the analysis and results. Next, the authors might consider moving the 'Methods' / 'Evaluation data' more up front, ahead of the model and site descriptions. This way, the descriptions could then focus on the model and site information most directly relevant to the specific evaluations conducted here. One example, as pointed out in a previous review comment, is that PFT information is not included for JSBACH – and for the other two apparently in the end are effectively just cold C3 grasses? Here's how this might look: rather than subsections for each model, it would read "here we evaluated model A (ref), B (ref) and C (ref). For process X, model A & B are similar in that they do Y, but

model C is different because it does Z". Again this puts the emphasis on the process and organizing it this way allows the reader to keep track of and more directly compare the models across the key evaluations presented in the results. Another potential addition to consider, at the beginning of the methods perhaps, is a short but explicit definition / overview of the scope of inference of the study. The scope has three angles, which follow the subsequent subsections of the methods, i.e. (1) indicators (evaluations included in this analysis, e.g. snow, ALT, soil C, CO2 fluxes, etc.), (2) processes (collectively included / compared among these 3 particular models), and (3) geography (climate, vegetation, permafrost etc. conditions across the 5 sites). Line 418: the suggestions above may help to expand on this statement about what / how C dynamics are intrinsically linked to the physical state of the system... 420-1: consider adding a sentence or two to point out the important, relevant results of the three previous studies referenced here. Did those results provide direction / motivation for this study? 469: the partitioning of CO2 vs CH4 is appropriately mentioned as a key issue here, but it is not evaluated or discussed elsewhere; do these models even simulate CH4 fluxes? 503: there is "very little carbon" – is that true when aggregated over the whole profile? Perhaps a 6th panel could be added to show / compare the total stocks simulated? 514-515: this implies that vegetation C via GPP is the only input, but would orchidee and jsbach also have inputs from thawed permafrost C that should be considered in the turnover rate?

Technical Corrections

I did not find any obvious technical errors in the text, but there are a couple of issues with the figures like inconsistent and/or missing labels for the site names in the mean and spread panels, lacking an explanation of what the error bars represent in Figure 4, and lacking explanation of what the dotted lines refer to in Figures 4 and 5.
* * *

---

## Referee Comment (RC3) · Anonymous Referee #2 · 25 Jul 2017

The idea would be to organize the paper as a whole around the arctic ecosystem processes that you are studying, so starting in the introduction with a description of each the key issue(s) with relevant background, uncertainty and hypothesis for this study. This formula could then be followed in the methods, as well (e.g. for each ecosystem process, how does each model represent it?).

I don't feel that a whole re-organization is that critical since the paper is already well-written. Just an idea... but I do think that a little more information in the introduction to set up the problem will help put this study in context for the reader, and give them some idea of what to expect when investigating model results with respect to these important

mechanisms.

---

## Short Comment (SC1) · 25 Jul 2017

Dear Reviewer,

Thanks a lot for your review! I have been looking through and I would just like clarification on one point. One of your very first comments (under the specific comments) reads: "...the 6 or so pieces serving as subsections of the results. More text on background and motivation could be added to the introduction for each, including more references for previous studies that have explored these issues."

I can't tell whether your suggestion would be to add some text on each aspect in the

overall introduction, or to add the text in the results section at the start of each subsection. I think that either could potentially work, but it would be great to know what you were thinking of!

Thanks and best wishes, Sarah

---

## Author Comment (AC3) · 9 Aug 2017

Thanks a lot for this clarification, that makes sense!

---

## Author Response (AR2)

**Response to reviews**

We thank the editor for their positive comments on the manuscript. We have addressed the comments from the editor's report as follows:

Typo at line 833: heterotrophic (add h) Done

Lines 863-864: Can you support this with a citation? Added reference to Palmer et al (2012) – see marked-up manuscript, below.

Line 879: Maybe use a different verb than showing (cf the use of shows earlier). Replaced 'showing' with 'demonstrating'

Lines 900-901: rewrite sentence. Sentence now reads: "Tundra vegetation should ideally be represented using several different PFT's, for example grasses and shrubs differ in carbon storage and their interactions with snow."

Following this page are the response to both reviewers, followed by the marked up manuscript and supplementary material.

Best wishes,
Sarah Chadburn (on behalf of co-authors)

Review 1

We thank the reviewer for their comments, and helpful ideas for improving the manuscript. Please find below our point-by-point response, and below this the proposed revised manuscript and additional supplementary material follows (the latter is titled 'Sensitivity to snow' and will be included with the supplementary material from the original submission).

Model description
How about a Table summing up the 3 models main features? This would allow significant text shortening. Also please make sure equivalent information is given for all 3 models. For example, vegetation details are lacking for JSBACH.
The model description section is now amalgamated into a single section, so that anything that is common to all 3 models is only mentioned once, and differences between models are explicitly pointed out. This way equivalent information is now given for each one. This has shortened the text. we have also added a table summarising the main model features, for clarity – see Table 1 in the marked up manuscript, following these comments.

By the way, PFT is defined nowhere and some institute abbreviations are not explained (IPSL, NCSDC). I let the editor decide whether that is necessary.
We have defined PFT and NCSCD. We have left the abbreviations for the Earth System Models as their full names are not relevant, but the editors can request that to be changed if necessary.

Lichens are not mentioned in any model description, from which I assume that they are not considered. Yet, they can be very abundant at some Arctic sites, sometimes covering most of the ground. They have physical and biological properties very different from mosses, for example a much lower thermal conductivity and different hydrological properties which strongly impacts the ground thermal and hydrological regimes. Please consider specifically mentioning this omission.
The moss model in JSBACH is actually designed to represent 'average' properties of bryophytes and lichens, rather than considering them separately. One reason for this is that the properties can vary more between species than they do between the phyla as a whole. In the model description we have added the following: "This model represents both mosses and lichens by one plant functional type with 'average' physiological properties." However as you suggest, it can be important to consider them separately and we have added a comment on this in the discussion (see below).

A couple of sentences or a line in the future model Table to describe the snow scheme would be nice (single layer, multilayer...). In fact all the models have a multilayer snow scheme. We included this in the re-written model description section and in the new table (see above).
Please also specify here that nutrient aspects are not treated in any of the 3 models. Done

Site description
The description of all sites should really be homogenized and considerably shortened, by at least 3 pages. What is in Table 1 need not be repeated in the text. Incomparable data are often given in the text. For example, some sites have mean annual temperature, others January and July, please be consistent.
We have shortened these to less than half a page for each site, and removed everything from the text that can also be found in the table. See marked-up manuscript following these comments.
Also detail the snow fraction of precipitation in all cases: "most" is vague and not very useful.
Snow fraction of precipitation can vary a lot from year to year at some of these sites, but nonetheless we have found some indicative values in the literature and added these to the table of climatic and permafrost variables.

All plant Latin names must be in italics. By the way, line 219, what are the Salix? Richardsonii, arctica, other? The Salix are usually Salix pulchra, which is a small (up to 50 cm high) willow shrub. We have checked all of the latin names in the revised version.

Field data

Measuring snow precipitation and snow depth in a reliable and representative manner is always a problem in the Arctic and the text does not convince me that this aspect was treated properly. Moreover, its impact may be understressed here since it conditions the permafrost thermal regime and therefore all carbon processes. How about details of the precipitation measurement, such as the presence of a wind shield around the gauge? I understand that precipitation measurements were not used, but since snow depth measurements are not convincing, as detailed below, perhaps analyzing precipitation data in more detail would be useful.

First of all, while precipitation measurements were not used for snowfall, they were used for rainfall, for which they are more reliable. This was not made clear in the manuscript so we have added a note: "However, the local precipitation measurements were still used for rainfall, as this is much more reliable, with an average undercatch of around 10% (Yang et al., 2005)."
At some sites there is no wind shield (e.g. Samoylov and Bayelva), and at others there is a wind shield (e.g. Abisko). We have added after the above line "(depending on the set-up of the precipitation gauge, which differs between sites)"
For snowfall, as discussed, the direct precipitation measurements are not reliable. However, to address the question about the impacts of the uncertainty in snowfall forcing, we have performed a sensitivity study and assessed its impact on the carbon-cycle processes – see below.

Was there any attempt to correct measured snow precipitation as described in (Forland et al., 1996)? This can double estimates of precipitation amounts and considerably improve agreement with snow accumulation.

Thanks for this suggestion. At several of these sites, the precipitation gauges installed do not actually detect snowfall, reducing the possibility to apply this in this study – along with the issue of wind-redistribution that reduces the correlation between precipitation and snow depth on the ground (for example, we found no correlation between snow depths at the Abisko mire and the nearby research station). However it is certainly a good idea to consider this approach for future studies. In general, for this study, we considered using the observed snow depths to be the best way of constraining snow precipitation for these sites, but additionally we have now performed a sensitivity study (see below) to show the impact of the potentially large uncertainties in snowfall forcing.

Measuring snow depth in a representative manner is difficult. Certainly using one point measurement is inadequate. In particular, in low-centered polygons, variations are huge and at least 100 measurements are required for a representative value. Please detail the representativity of your snow depth measurements. In case the data are found to have limited representativity, this should be clearly stated and perhaps a sensitivity study would be useful (if it is still possible to perform it): what is the impact of snow amount on permafrost temperature and carbon cycling?

The snow depth measurements are point measurements for most sites, except for Abisko, where measurements are averaged from several locations on the mire. Considering the representativity for each site: The Abisko measurements are deliberately taken to give a representative sample. At Zackenberg, there is a CALM grid at the site where snow depths are measured periodically. Snow depths are relatively homogeneous here and the point observation appears to be representative. At Samoylov and Kytalyk there will be variability due to polygon structures and wind distribution as you suggested, so in general point observations are not representative (see also the comparison with GlobSnow snow water equivalent, below – here it appears that the Samoylov simulation matches better with the GlobSnow product than Kytalyk.) Finally at Bayelva, it is hilly and there can be some variation in the flux tower footprint, so the point observation may also not be representative for this site, in fact it seems to be a little higher than the 'typical' values. We have added a full discussion of the representativity of snow depths in the supplementary material. We have then

performed a sensitivity study with two of the models for all of the sites, which aims to cover all uncertainties including where the single snow depth measurement was not deemed representative of the flux tower area. The runs were repeated twice with snowfall increased by 50% and reduced by 50%, respectively. We added the details of this in the supplement and some discussion in the text. In general there can be significant differences in the carbon cycling, in particular for JULES – this is because the snow impacts the soil moisture availability. For two of the sites (Kytalyk and Samoylov) this resulted in very different vegetation fractions during spinup and therefore a big difference in soil carbon stores. For JSBACH, however, the differences are fairly minimal. It is clear that the differences in GPP and Reco are due to soil moisture in JULES as the vegetation only responds to soil moisture and climate forcing in the model, and we see clearly the same patterns in all these variables:

[Figure]

As expected, all sites show an overall warming of the soil due to increase in snow depth, with the majority of the warming in winter. In JSBACH this can also be seen to impact the soil carbon stocks (following figure). In JULES, the impact of vegetation differences on soil carbon is larger than the impact of warming and dominates the changes.

[Figure]

A full discussion of the sensitivity study, with plots, is added in the supplementary material (also included following these comments). In the main text we have added some discussion in the section on snow, regarding the poor simulation for Abisko (see below), and the following: "It is important

to be careful when modelling snow depth based on single point observations, as they may not be representative of the area as a whole. Further details on the representativity of snow depths are given in supplementary information. The sensitivity of carbon cycle processes to increased/reduced snowfall is discussed in Sections 3.5 and 3.6.1."

We then include discussion in Sections 3.6.1 (see below) and 3.5: "The soil carbon stocks are sensitive to changes in snow depth in these models (see supplementary Figure S8), through changes in soil temperature (JSBACH) and changes in vegetation growth (JULES). In JULES, both vegetation and soil temperature changes affect the soil carbon, but the vegetation effect dominates. In fact, for two of the sites (Kytalyk and Samoylov), the vegetation coverage is so different during spinup that the simulation with increased snowfall accumulates twice as much soil carbon as the default case (although the stocks are still much too small and the absolute difference is less than 10 kgm$^{-2}$ in the whole soil column)."

A note about the sensitivity study is also added in the methods: "Even with these corrections, there is still considerable uncertainty in precipitation forcing, particularly the snowfall, so in order to test the impact of this, two of the models (JULES and JSBACH) performed two additional sets of simulations, with snowfall increased and reduced by 50%."

Perhaps looking at data from reanalyses would also be helpful for an extra evaluation of precipitation and snow depth data.

Since the snow depth is not controlled by precipitation at many of these sites, but much more by the wind, we decided not to look at any more precipitation data. However, the Globsnow reanalysis data could be useful to compare against SWE in the models. Unfortunately, for Zackenberg (Greenland) and Bayelva (Svalbard), there are no values in the dataset as these are very small pieces of land between glaciers and ocean. There is also no value given for the closest pixel to Abisko (which may be because the site is next to a lake?). Taking the next pixel along gives an SWE that is much higher than could be expected for the mire site, which is not surprising given the landscape is mountainous and snow depth will be very variable around this region – however, this precludes using Globsnow data for Abisko. This leaves two sites: Kytalyk and Samoylov, which are flatter and more homogeneous landscapes where the product should be more representative: Langer et al. (2013) showed that the globsnow SWE data matched well with the Samoylov island data assuming constant snow density of 250 kg/m$^3$. We can compare directly with modelled SWE. Since we can only do this for 2 out of 5 sites this does not merit an extra figure in the text but we include a plot here (showing average of 2005-2013):

[Figure]

This shows a reasonable simulation of SWE for Samoylov but too little for Kytalyk (despite the models matching snow depth quite closely). This may be because there is a larger uncertainty in Kytalyk snow depth, due to having a limited number of years of in situ measurements, or because the snow is more compacted in reality than the models, or alternatively because the point measurement is not representative of the larger area. The GlobSnow product also varies in accuracy depending on proximity to ground stations. Further investigation would be required to confirm the

reason for the discrepancy.

Are there any field measurements of snow density to validate model assumptions of this variable?
There are field measurements or literature values available from some of the sites. We can also output this from the models. In JSBACH the snow density does not vary much between sites, whereas for JULES and ORCHIDEE, density varies more between sites, but is quite consistent between these two models, suggesting that they are constructed similarly. At Samoylov, the estimated density is between 200 and 400 kgm$^{-3}$ (20$^{th}$ April) depending on the type of snow (Gouttevin et al., 2017), whereas JULES and ORCHIDEE simulate a lower density (around 180 kgm$^{-3}$). The work of Gouttevin et al. (2017) suggests that the reason for this difference is likely because the models do not simulate wind compaction. Similarly at Zackenberg, the average density for April-May is high (around 375 kgm$^{-3}$, https://data.g-e-m.dk), and the models simulate a lower density (270-350 kgm$^{-3}$). On the other hand, at Bayelva the mid-season snow density is 305 kgm$^{-3}$ (Gisnås et al., 2014), which is very close to the values simulated in JULES and ORCHIDEE. In the Supplementary material we have added a comment on this: "It is also useful to compare snow density in models and observations. For example, recent work shows that including wind compaction is essential to capture high snow density at Samoylov (Gouttevin et al., 2017), and indeed our models show a snow density closer to the 'default' models in Gouttevin et al. (2017), which is too low due to the omission of wind compaction processes."
It would make an interesting study focussing on the snow dynamics in these models at these sites – we hope that by collating all of the data for this study we have opened up opportunities for further detailed studies. We have added a comment on this at the end of the conclusion: "This work also opens up opportunities for further process studies in future."

By the way, snow temperature measurements at several heights can be very useful to evaluate the validity of snow schemes, and implementing those at the sites described here may be valuable for future work (Barrere et al., 2017).
We have added a comment on this in the supplementary material: "It is also important for the models to better represent the profile of snow thermal conductivity: for example the models do not simulate the low-conductivity 'depth-hoar' layer that can form at the base of the snowpack (Domine, et al. 2016). For this, monitoring of snow temperature at different heights can be valuable to improve the models (Barrere et al., 2017)."
Along with further comments on the need for better representation of snow in these models.

Lines 327 and 329: please use "snow depth" throughout.
Done

Results and discussion
Lines 431-432. The snow depth model output at Abisko is not "reasonable". It just does not seem to work there. Please consider representativity of field data and modify discussion.
The snow depth measurements for Abisko are actually an average taken from several locations on the mire and are representative for snow depth on the mire. The representativity of the snow data is now discussed in detail in the supplementary material. A more realistic simulation for Abisko has now been made in the -50% snow case, in our snow sensitivity study. We showed that, in general, the models are sensitive to changes in snow. However at Abisko the carbon stocks/fluxes do not show major changes. We have added a comment in the snow results/discussion section: "...for the most part the models make a reasonable simulation of the snowpack accumulation and compaction, with the exception of Abisko where the models are all biased high. Here, snow inputs are particularly uncertain as no high-resolution timeseries of snow depth are available (unlike the other sites). We performed a sensitivity study to test the impact of uncertainties or variability in snow depth on the simulated carbon-cycle processes. In this study, a reduction of 50% in snowfall allows the models to simulate a realistic snow depth at Abisko – see supplementary material. The impacts

on soil carbon stocks and fluxes are fairly small, however (between 0.2% and 10%, supplementary Figures S7 and S8)."

Line 433. "snow often melting a little too early" in simulations. Ambiguous as written.
This sentence is re-written as: "During the melting season the models are less accurate than during accumulation, with the snow often melting too early - by up to 25 days in the most extreme case."

Lines 436-437. How do models account for vegetation effects on snow albedo?
Snow albedo is reduced by the presence of vegetation, more so when the snow is shallower or the vegetation is taller. (For example, in JULES, the albedo is interpolated between the snow albedo and the snow-free albedo according to snow depth, d, and the vegetation roughness length, $z_o$ (Essery et al 2003): snow-covered fraction = d / (d + 10$z_o$) )
In the text we have added: "(this is modelled by interpolating between snow-covered and snow-free albedo depending on snow depth and vegetation height)"

Lines 457-458. How about thermal conductivity values obtained by JULES, and how do they compare with other models? Perhaps also compare with values obtained at a comparable high Arctic sites in low-centered polygons (Domine et al., 2016) if you think this supports your case. Note by the way that stratification of thermal conductivity can have an important effect, as suggested by (Barrere et al., 2017), so that one-layer snow models can give the correct mean thermal conductivity value while making a large error on atmosphere-ground heat fluxes. Incorrect snow thermal conductivity stratification can also lead to incorrect timing of ground freezing and thawing. Arctic snow often has a very low thermal conductivity layer at the base, which delays freezing and thawing. This process is missed if the snow scheme gives a high thermal conductivity to the basal snow layer.
Thanks for this, it was helpful to compare the values in JULES with these observations. We have added in the main text: "Indeed, the conductivity of snow in the JULES simulations is between 0.03-0.1 $Wm^{-1}K^{-1}$ at the sites with shallow snow (and in the upper layers of the snowpack at sites with deeper snow), which is considerably lower than typical values for similar tundra sites, which suggest a realistic conductivity would be around 0.2-0.3 $Wm^{-1}K^{-1}$, at least for the upper part of the snowpack (Gouttevin et al., 2012b; Domine et al., 2016)."
We have also added a comment about the need to represent a low conductivity layer at the base of the snow in the supplementary discussion (see above).

Line 559. A word on nutrients here?
We have added this, so this part now reads:
"Of these, climate is the main driver of vegetation growth in these models (since nutrient limitation is not included, the soil only impacts the vegetation through moisture stress...)"

Line 574-575. "GPP depends mostly [...]on shortwave radiation in the second half of the season". How about moisture? For example, (Frost and Epstein, 2014) stated that "rates of shrub [...] expansion were not strongly correlated with temperature trends and were better correlated with mean annual precipitation".
This is a good point, vegetation growth is correlated with soil moisture, and our sensitivity study with increased/decreased snow depth has confirmed this in the JULES model. However (as your quote implies), the moisture effect occurs over longer timescales than daily and hourly variability, which is driven more by shortwave radiation. The sentence in the manuscript will be better phrased as "In particular, the increase in GPP in the first half of the season is driven by increasing LAI, and the downward trend of GPP in the second half of the season is driven by shortwave radiation".
We have then added a comment about the impacts of soil moisture (end of Section 3.6.1):
"Carbon fluxes are also sensitive to soil moisture, as seen in simulations with increased/decreased snowfall, where differences in soil moisture availability in summer are reflected by changes in

annual mean GPP, ecosystem respiration and vegetation fraction in JULES (Supplementary Figure S7), in line with Frost and Epstein (2014). Therefore, realistic simulation of precipitation and soil moisture is a pre-requisite for improved LAI and vegetation dynamics."
We have also added a sentence in the conclusion: "There is also a need to address remaining issues in the model physics, particularly for soil moisture and snow."

Conclusion
The impact of mosses is stressed, but as mentioned above, I really think that lichens can have a huge impact. I gather that they are not very important at the sites studied here, but on a pan-Arctic scale, this is probably different.
We agree, and we have added in the discussion: "It could also be important to consider lichens separately from mosses, as their physical and biological properties can be very different. For example, the high albedo of lichens can impact the Earth's radiation budget (Bernier et al., 2011)."

Since you are talking about landscape dynamics, you may talk about the impact of lakes and ponds caused by landscape dynamics such as thermokarst lakes formation. These lakes are often hotspots of GHG emissions. See e.g. (Bouchard et al., 2015) and references therein.
We agree this is a significant omission from our models on a large scale. We have added a note on this in the conclusion: "Lakes and ponds also play a major role in methane and carbon dioxide exchange with the atmosphere (Bouchard et al., 2015; Langer et al., 2015) and should also be considered in future land surface models."

Figure1. What is the meaning of mean snow depth? Spatial mean? Temporal mean? Over what period? The Abisko graph does not seem to match the mean value.
The plots show 'mean annual cycle', and this means we take the average for a given time of year across a number of years observations – but generally for a single site. The years used for every site and for every variable (including snow) are given in the supplementary information and this is referenced in the captions for all the figures. In fact, Abisko is slightly different from the other sites in that the observations have very low temporal resolution, but they are in fact averaged across different locations on the mire and therefore are representative for the area as a whole. On the figure caption, we have clarified: "Mean annual cycle is calculated from a single site over a number of years, except for Abisko where measurements were taken in several different locations on the mire."
Regarding the mis-match between the models and observations at Abisko, we have added some discussion on this – see above.

Table 1: What is summer? What is winter? Permafrost T, at what depth?
Permafrost temperature is not measured at a consistent depth at the sites, these are approximate values anywhere below the active layer but they give an indication of the differences in permafrost conditions between sites. Summer and winter generally refer to the maximum and minimum average monthly temperatures. We have clarified this in the table by changing "summer" to "max. monthly" and "winter" to "min monthly".

We thank the referee very much for their helpful review. We have addressed all of the issues raised and give details of this below, with the proposed revised manuscript following the comments.

While the study is laid out well in the paper, I do wonder if some re-structuring and/or additional information could strengthen the emphasis on the key ecosystem processes being studied here, i.e. the 6 or so pieces serving as subsections of the results. More text on background and motivation could be added to the introduction for each, including more references for previous studies that have explored these issues. Perhaps it'd be useful in the introduction to add a sentence on each issue that frames them as questions or hypotheses as a preview of what the reader can expect to see in the analysis and results.
We have added such a 'preview' in the introduction (penultimate paragraph), which references some relevant studies and outlines what will be addressed in our study. This paragraph now reads as follows:
"In this paper we assess the ability of the land surface components from three Earth System Models to represent the observed carbon stocks and fluxes at tundra sites, identifying the processes that have the greatest impact on the uncertainty. These processes are therefore priorities for future model development. Observational studies in tundra environments have shown that carbon dynamics are sensitive to physical conditions (Lund et al., 2012; Cannone et al., 2016; Pirk et al., 2017), so we first assess the ability of the models to capture the mean physical state of the system and the differences between sites, specifically in terms of snow depth, soil temperature, soil moisture and active layer depth. Secondly, soil carbon stocks are evaluated against measured soil carbon profiles, assessing the main causes of biases in the models. Half-hourly NEE data from eddy flux towers are used to evaluate the simulated carbon fluxes, comparing the models directly against observations before analysing the relationships between ecosystem carbon fluxes and different driving variables. We also consider the impacts of other controlling factors such as nutrient limitation and mosses, whose importance has been identified in previous studies (Atkin, 1996; Uchida et al., 2009)."

Next, the authors might consider moving the 'Methods' / 'Evaluation data' more up front, ahead of the model and site descriptions. This way, the descriptions could then focus on the model and site information most directly relevant to the specific evaluations conducted here.
We have moved the evaluation data section to go first in the methods, followed by model description, then site description, and finally the simulation set-up. The Methods section is also now introduced as you suggested, see below*.

One example, as pointed out in a previous review comment, is that PFT information is not included for JSBACH – and for the other two apparently in the end are effectively just cold C3 grasses? Here's how this might look: rather than subsections for each model, it would read "here we evaluated model A (ref), B (ref) and C (ref). For process X, model A & B are similar in that they do Y, but model C is different because it does Z". Again this puts the emphasis on the process and organizing it this way allows the reader to keep track of and more directly compare the models across the key evaluations presented in the results.
We have completely re-written the model description section following your suggestions. This can be found in the marked-up manuscript attached at the end of this file.

Another potential addition to consider, at the beginning of the methods perhaps, is a short but explicit definition / overview of the scope of inference of the study. The scope has three angles, which follow the subsequent subsections of the methods, i.e. (1) indicators (evaluations included in this analysis, e.g. snow, ALT, soil C, CO2 fluxes, etc.), (2) processes (collectively included / compared among these 3 particular models), and (3) geography (climate, vegetation, permafrost etc. conditions across the 5 sites).
*At the beginning of the Methods section we have added: "This study takes three different angles: 1) Comparison with observed indicators. 2) Comparison of processes between models. 3) Comparison of geographical conditions (e.g. vegetation, permafrost) between sites. The structure of the methods section follows this, describing firstly the observational indicators used (Section 2.1), secondly the processes represented in the models (Section 2.2), and thirdly the conditions at the sites (Section 2.3). Lastly, details of the simulation set-up and forcing data are given in Section 2.4."

Line 418: the suggestions above may help to expand on this statement about what / how C dynamics are

intrinsically linked to the physical state of the system...

We have expanded this statement to give an example of the links between C dynamics and physics, so it now reads: "The carbon dynamics are intrinsically linked to the physical state of the system (for example, determining the rate of soil carbon decomposition), so we start by assessing the snowpack, soil temperature, soil moisture, and active layer thickness in all three models."

420-1: consider adding a sentence or two to point out the important, relevant results of the three previous studies referenced here. Did those results provide direction / motivation for this study?

We have added the following: "In these studies, representing organic soil was identified as a key influence on the simulation of soil physics, and following this we compare organic against mineral soils in our analysis."

469: the partitioning of CO2 vs CH4 is appropriately mentioned as a key issue here, but it is not evaluated or discussed elsewhere; do these models even simulate CH4 fluxes?

Although this is of course a key issue, we did not have the model capability to simulate CH4 in this present work. However, we certainly shouldn't have ignored it in our manuscript! Accordingly, we have added the following paragraph in the conclusion:

"The feedbacks between the Arctic and the global climate are strongly dependent on whether carbon is released into the atmosphere from heterotrophic respiration as carbon dioxide or methane. The modelling capability at the time of this study was not sufficient to simulate the methane flux. However, this development is in progress, see e.g. Kaiser et al. (2017), and represents an important topic for future work."

503: there is "very little carbon" – is that true when aggregated over the whole profile? Perhaps a 6th panel could be added to show / compare the total stocks simulated?

As suggested we have added a 6th panel to the plot. This shows the total column and top 1m soil carbon for the default versions of JULES and ORCHIDEE (it was too crowded to also show the experiments: 'jules fixed veg' and 'orchidee no mix', and line 503 refers to the default versions). The observations for the top 1m are also shown. We have made the statement on line 503 less ambiguous. It now reads: "The most obvious problem is underestimation: there is much too little carbon simulated at many of the sites (see last panel on Figure 5)."

514-515: this implies that vegetation C via GPP is the only input, but would orchidee and jsbach also have inputs from thawed permafrost C that should be considered in the turnover rate?

Thank you for this important comment. In fact, JULES and ORCHIDEE model versions used in this study do represent a soil carbon mixing process into the permafrost, while the current JSBACH model does not. This affects the effective mean residence time. The important process is that some of the carbon from primary production has been transported into the permafrost over decades to centuries, and this way excluded from the carbon cycling. Such effects are represented implicitly by a box model approach like the one used in JSBACH, due to the effective soil temperature and its effect on carbon decomposition.

With climate change and permafrost thawing, this carbon can enter the cycling again, and be decomposed by microbes. However, that was not the question here. With Fig. 6 we address the question of the validity of turnover processes: How much of the carbon from primary production will stay in the soil over a long period of time, during pre-industrial conditions? For example, ORCHIDEE results suggest that (even when considering soil carbon mixing) carbon turnover rates are far too high, since GPP is reasonable but soil carbon stocks are underestimated. The opposite is true for JSBACH results at Zackenberg. In contrast, even when some of the JSBACH and JULES GPP and carbon stock results do not match perfectly with the observations, turnover processes seem to be better parameterized, e.g. in case of JSBACH and Kytalyk. Higher GPP results would also lead to higher soil carbon stock results here.

Technical Corrections

I did not find any obvious technical errors in the text, but there are a couple of issues with the figures like inconsistent and/or missing labels for the site names in the mean and spread panels, lacking an explanation of what the error bars represent in Figure 4, and lacking explanation of what the dotted lines refer to in Figures 4 and 5.

Where the labels are inconsistent or missing in the mean/spread panels, we have changed them so that they all take the same form: Abisko, Bayelva, Kytalyk, Sam., Zack. And just to be completely clear we have added the following in the captions: "
[revised manuscript text omitted]

**Sensitivity to snow**

**Representativeness of snow depths**

In flat, open tundra landscapes, the snow is heavily affected by wind blowing, with the consequence that snow depth does not correspond directly to precipitation, and therefore using direct snowfall measurements is not possible in these landscapes. This scenario particularly applies to Kytalyk, Samoylov, and the Abisko mire. On a large scale, the snow can be quite even distributed due to the flat landscape (e.g. Blok et al., 2010, Table 2), but the microtopography at these sites (e.g. ice wedge polygons, palsas) leads to small-scale variability in snow depths. For example at Samoylov the depressed polygon centers have much deeper snow than the elevated rims (Boike et al., 2013). Thus a single point measurement of snow depth may not be representative of the whole flux tower footprint. At Abisko, however, several locations on the mire are averaged to give a representative sample.

At Bayelva and Zackenberg, the landscape is more mountainous, and there is more variation in snow depth around the area due to the topography of the land (and consequent differences in vegetation). At Zackenberg the snow is measured on transects across different vegetation types and the values range from snow-free to more than 1m of snow at a single time. However, the flux tower is situated in a fairly homogenous *cassiope* heath where snow surveys show the typical standard deviation of snow depth any one point in time is around +/-12cm (ZEROCALM1, https://data.g-e-m.dk/, average depth around 50cm). For this site, the point observation appears to be representative of the flux tower footprint. At Bayelva the snow depth varies by around +/-50% within the vicinity of the flux tower (Gisnås et al., 2014), and our point observation falls a little higher than the typical values for maximum snow height.

Even for sites where a point measurement of snow depth is representative of the flux tower area, the snowfall timeseries is derived using an assumed density and could be better parameterised using snow density measurements.

**Sensitivity study**

To investigate the impact of the variability and uncertainty in snow depth, we performed a sensitivity study. The observations suggest that increasing and decreasing the snow depth by 50% from the model simulated values would capture the range of observed snow depths in each of these landscapes. Since the snow depth is dynamically simulated rather than input to the models, we approximated the change in snow depth by increasing and decreasing the snowfall forcing by 50%. Two of the models (JSBACH and JULES) were then re-run (including spin-up) in these two different configurations. Snow depth in these simulations now spans a range that includes the point observations (Figure S5).

As expected, increased snow depth leads to an overall warming of the soil for every site, and reduced snow depth leads to a cooling (Figure S6). However, most of the change happens in winter, where it will have less impact on the carbon cycle since the vegetation and soil decomposition processes take place mainly in summer (JJA) (Figure S6).

[Figure]

**Figure S5 |** Mean annual cycles of snow depth (as in Fig. 1 in main manuscript) showing simulations with increased and reduced snowfall in JSBACH and JULES.

[Figure]

**Figure S6** | Mean soil temperature in different seaons, showing simulations with increased and reduced snow for JULES and JSBACH. (DJF=December, January, February. MAM=March, April, May. JJA=June, July, August. SON = September, October, November.)

Vegetation growth is not directly impacted by snow or soil temperature changes in these models. However, the change in winter snowfall also leads to changes in soil moisture during summer, which does affect vegetation growth. An increase in snow should lead to an increase in water infiltration into the soil in spring and thus an increase in the available soil moisture. In JULES, however, for two of the sites (Zackenberg and Bayelva) the opposite effect is seen, where increased snow depth leads to less soil moisture in summer, and vice versa (Figure S7). In JULES, the changes in soil moisture are reflected in the GPP, ecosystem respiration (Reco) and vegetated fraction, which all increase with higher soil moisture and reduce with lower soil moisture (Figure S7). At many of the sites these are significant changes (although they still leave the model with low values of GPP/Reco compared to observed fluxes). The impact of any change in GPP is amplified by the resulting changes in vegetation fraction. In JSBACH, however, the changes in soil moisture, GPP and Reco are not significant (Figure S7).

Soil carbon stocks are impacted directly by the soil thermal state (as well as soil moisture, and inputs from vegetation). For JSBACH, while the vegetation fluxes do not show any noticeable sensitivity to snowfall (Figure S7), the soil carbon has a small but consistent trend towards lower soil carbon in the simulations with increased snow (Figure S8), which – since the other influencing variables have not significantly changed – is most likely due to consistently higher soil temperatures when more snow is present. For JULES, however, any changes in decomposition due to soil temperature are obscured by larger differences of vegetation inputs, particularly for Kytalyk and Samoylov sites (Figure S8), where the vegetation fractions are very different during spinup for the different sensitivity tests, and thus the rate of soil carbon accumulation changes significantly.

[Figure]

**Figure S7 |** Impacts of increased/reduced snowfall on soil moisture and carbon-cycle related variables (GPP, ecosystem respiration, and vegetated fraction), in JSBACH and JULES.

[Figure]

**Figure S8 |** Impact of increased/reduced snowfall on soil carbon stocks in JSBACH and JULES.

**Discussion**

Our sensitivity study has shown a high sensitivity of surface soil temperature to a 50% change in snow depth of up to 5°C or more, seasonally. This is in line with observations (Gisnås et al., 2014). Soil carbon decomposition is sensitive to these soil temperature changes, resulting in lower carbon stores for the warmer simulations in JSBACH (Fig. S8), which is in line with studies such as Lund et al. (2012) which showed that snow affected the carbon budget at Zackenberg by warming the soil and increasing soil respiration. However, the impact of snow on soil moisture is not in line with observed behaviour: in general, more snow should lead to increased soil moisture availability in summer (see for example Litaor et al., 2008). However, in JULES for two of the sites, the summer soil moisture is reduced with additional snowfall, and in JSBACH there are no significant changes. This supports the conclusion that more work is needed on the hydrology schemes in these models. Furthermore, the models are missing some snow-vegetation interactions such as preventing vegetation growth when covered by snow, or protection from damage in winter.

It is also important for the models to better represent the profile of snow thermal conductivity: for example the models do not simulate the low-conductivity 'depth-hoar' layer that can form at the base of the snowpack (Domine et al., 2016). For this, monitoring of snow temperature at different heights can be valuable to improve the models (Barrere et al., 2017). It is also useful to compare snow density in models and observations. For example, recent work shows that including wind compaction is essential to capture high snow density at Samoylov (Gouttevin et al., 2017), and indeed our models show a snow density closer to the 'default' model in Gouttevin et al. (2017), which is too low due to omission of wind effects.

In large-scale modelling, it is certainly important to represent variability in snow depth, which is only coarsely included in land surface models in most cases (e.g. snow depth varies only between surface tiles (Essery et al., 2003)). For recent developments towards this, see for example Gisnås et al. (2014).